# The poxvirus F17 protein counteracts mitochondrially orchestrated antiviral responses

Nathan Meade [1], Helen K. Toreev[1], Ram P. Chakrabarty [2], Charles R. Hesser[1], Chorong Park[1], Navdeep S. Chandel [2] & Derek Walsh [1]✉

Poxviruses are unusual DNA viruses that replicate in the cytoplasm. To do so, they encode approximately 100 immunomodulatory proteins that counteract cytosolic nucleic acid sensors such as cGAMP synthase (cGAS) along with several other antiviral response pathways. Yet most of these immunomodulators are expressed very early in infection while many are variable host range determinants, and significant gaps remain in our understanding of poxvirus sensing and evasion strategies. Here, we show that after infection is established, subsequent progression of the viral lifecycle is sensed through specific changes to mitochondria that coordinate distinct aspects of the antiviral response. Unlike other viruses that cause extensive mitochondrial damage, poxviruses sustain key mitochondrial functions including membrane potential and respiration while reducing reactive oxygen species that drive inflammation. However, poxvirus replication induces mitochondrial hyperfusion that independently controls the release of mitochondrial DNA (mtDNA) to prime nucleic acid sensors and enables an increase in glycolysis that is necessary to support interferon stimulated gene (ISG) production. To counter this, the poxvirus F17 protein localizes to mitochondria and dysregulates mTOR to simultaneously destabilize cGAS and block increases in glycolysis. Our findings reveal how the poxvirus F17 protein disarms specific mitochondrially orchestrated responses to later stages of poxvirus replication.

The *poxviridae* are a family of large double-stranded DNA (dsDNA) viruses that include Variola Virus (VarV) and Monkeypox Virus (MPXV), the causative agents of smallpox and Mpox diseases, respectively[1,2]. Smallpox was eradicated through vaccination with live and attenuated forms of Vaccinia Virus (VacV), a closely related *Orthopoxvirus* that also serves as the laboratory prototype for poxvirus research. Poxviruses are unusual amongst mammalian DNA viruses in that they are highly self-sufficient and replicate entirely in the cytoplasm, forming DNA-filled structures termed viral factories (VFs)[1]. However, their mode of replication leaves them highly vulnerable to cytosolic DNA sensors, in

particular cGAMP synthase (cGAS)[3,4]. Upon binding to DNA in the cytoplasm, cGAS produces 2′,3′-Cyclic GMP-AMP (cGAMP) to activate Stimulator of Interferon Genes (STING), which subsequently translocates from the endoplasmic reticulum to the Golgi network to activate TANK Binding Kinase (TBK1). Activated TBK1 in turn phosphorylates Interferon Regulatory Factor 3 (IRF3) and nuclear factor-kB (NF-kB), which translocate to the nucleus and induce expression of interferon-stimulated genes (ISGs). In order to replicate, poxviruses encode approximately 100 immunomodulatory proteins that counteract cGAS and several other antiviral sensing and response pathways[4]. Notably,

[1]Department of Microbiology-Immunology, Feinberg School of Medicine, Northwestern University, Chicago, IL 60611, USA. [2]Department of Medicine, and Department of Biochemistry and Molecular Genetics, Feinberg School of Medicine, Northwestern University, Chicago, IL 60611, USA.
✉e-mail: derek.walsh@northwestern.edu

many of these immunomodulators are variable host range determinants and virtually all are expressed very soon after viral entry to rapidly disarm antiviral responses and help establish infection. However, significant gaps remain in our understanding of poxvirus sensing and evasion strategies throughout the course of their replication cycle.

We recently found that the late viral protein, F17 functions to block cGAS-dependent ISG responses at later stages of infection[5,6]. F17 is a structural component of the poxvirus core that is required for the morphogenesis of newly formed virions into mature infectious particles[7,8]. Because of this function, a virus that expresses F17 in an isopropylthio-β-galactosidase (IPTG)-inducible manner, termed iF17, is used to propagate stocks and study F17's roles in infection[7]. Virion-associated F17 is degraded after virus particles have entered the cell and uncoated, meaning that the iF17 virus behaves as a null mutant at later stages of infection in the absence of IPTG to induce de novo F17 synthesis[5-9]. Combined with construction of a revertant virus, termed iF17R, which restores F17 expression and rescues iF17 phenotypes[6], we discovered that beyond its role in virion morphogenesis, F17 also functions to evade antiviral responses by dysregulating mammalian/mechanistic Target of Rapamycin (mTOR)[5,6]. Moreover, phosphorylation at Serine 53 and 62 regulates F17's ability to target mTOR[5,6] but is not required for its role in virion morphogenesis[7,8], suggesting that phosphorylation controls the partitioning of F17's dual functions during infection.

While other viruses indirectly activate or repress mTOR through upstream regulators or downstream effectors[10], F17 directly targets and dysregulates mTOR complexes by binding and displacing their regulatory subunits, Raptor and Rictor[5]. mTOR is a multi-subunit metabolic sensor and effector kinase that adjusts the output of several cellular processes in response to mitogenic signals or the availability of various amino acids, energy sources or metabolites[11]. It is perhaps most widely studied for its roles in regulating cap-dependent translation initiation but mTOR also regulates other processes including autophagy, glucose metabolism and lipid synthesis[12]. Notably, the timing of F17 expression and resulting mTOR dysregulation coincides with a switch in viral translation initiation strategies from cap-dependent to cap-independent modes[13], suggesting mTOR dysregulation may serve to disrupt cellular processes that require precise control over mTOR. Indeed, wildtype (WT) or iF17R viruses that express F17 induce destabilization of a subpopulation of cGAS and do not induce ISG responses. By contrast, in the absence of F17, host cells retain normal control over mTOR activity, which is required to sustain cGAS stability and mount a robust ISG response to infection[5,6]. However, beyond contributing to cGAS destabilization, the nature of the antiviral responses that are blocked by F17 and why poxviruses target mTOR in this way remain unknown.

Here, we show that F17 dysregulates mTOR to block host antiviral responses to later stages of poxvirus replication that are driven by mitochondrial hyperfusion, which coordinates the release of mtDNA to prime cGAS together with an increase in glycolysis that supports ISG production.

## Results

### Mitochondrial DNA release stimulates ISG responses in the absence of F17

To better understand how F17 might function to counteract innate responses to infection, we first examined the localization of phosphorylated F17. As we have been unable to generate an F17 antibody suitable for immunofluorescence, and to specifically determine the localization of phosphorylated F17, we generated primary Normal Human Dermal Fibroblasts (NHDFs) expressing Flag control peptide or Flag-tagged phosphomimetic F17-S53/62E. Imaging showed that phosphorylated F17 predominantly localized to mitochondria in uninfected cells and to mitochondria which aggregated into hyperfused structures around the Golgi in cells infected with either WT, iF17

or iF17R viruses (Fig. 1a and Supplementary Fig. 1a, b). Mitochondrial fractionation approaches[14] provided independent evidence that F17 associates with mitochondria during infection (Supplementary Fig. 2). By contrast, imaging showed that a Flag-tagged non-phosphorylatable form of F17 (F17-S53/62 A) accumulated predominantly at VFs (Supplementary Fig. 3). This aligns with reports that F17 phosphorylation is not required for its function as a structural protein[7,8] and suggests that phosphorylation targets a subpopulation of F17 to mitochondria. While future studies are required to determine precisely how F17's mitochondrial localization is regulated, these findings provided the first indication that F17 might counteract host responses that are controlled by mitochondria.

Although poxviruses are known to sustain mitochondrial functionality[4,15], there is growing evidence that a number of viruses can cause leakage of mitochondrial DNA (mtDNA) into the cytosol which in turn activates cGAS[16-29]. However, whether mtDNA plays a role in responses to poxvirus infection remains unclear. In studies predominantly focused on herpes simplex virus type 1 (HSV-1), VacV was shown not to induce mtDNA release[22]. However, samples were only examined as far as 6 h post infection (h.p.i.) of primary mouse embryo fibroblasts, which represents a relatively early phase in VacV replication which spans 24–48 h in biologically relevant cell types (for examples, see[5,30]). By contrast, a recent study of Measles Virus (MeV) reported that Modified VacV Ankara (MVA) also induces mtDNA release and immune activation at 24 h.p.i.[31]. MVA is a highly attenuated strain that is missing many genes and expresses late gene products inefficiently in most cell types, leaving fundamental questions as to whether this is unique to MVA and how poxviruses might counteract mtDNA-driven responses if they occur naturally. To address these issues, we infected NHDFs with WT VacV for 6 h or 24 h. Staining samples with anti-DNA antibody that robustly stains mtDNA[18,22], we detected normal mtDNA nucleoids inside mitochondria in uninfected cells and in cells infected for 6 h (Fig. 1b, c). Notably, at this earlier timepoint only relatively small, early VFs were detectable and mitochondria had not yet begun to aggregate. However, by 24 h.p.i., when large VFs and hyperfused mitochondria had formed, fewer nucleoids were detectable inside mitochondria and instead, nucleoids were observed outside of mitochondria that had accumulated around the Golgi (Fig. 1b, c and Supplementary Fig. 4a).

While this established that WT VacV induces mtDNA release, VFs also stained with anti-DNA antibody and prevented us from accurately quantifying mtDNA release using this approach. As such, we fractionated cells and measured mtDNA levels in the cytosol by PCR and Real Time (RT)-qPCR[18,22]. PCR analysis confirmed that infection of NHDFs with either WT, iF17 or iF17R viruses increased the levels of cytosolic mtDNA, while mtDNA was degraded in HSV-1 infected cells as previously shown by others[22] (Fig. 2a). Similar release of mtDNA was observed upon infection of THP1 monocytes (Supplementary Fig. 4b). RT-qPCR analysis at different time points further showed that although not yet statistically significant, cytosolic mtDNA became detectable in infected cells by 12 h.p.i. and that these levels significantly increased through 24 h.p.i. (Fig. 2b). Notably, we detected modest reductions in viral DNA replication in iF17-infected cells (Supplementary Fig. 5), despite no detectable differences in protein production across different time points or kinetic classes of genes (Supplementary Fig. 6). This may reflect a need to dysregulate mTOR to control nucleotide pools or negative effects of the host antiviral response on viral DNA replication. Regardless of the underlying cause, there was no increase in viral DNA replication or viral DNA levels in the cytosol that might explain the increase in ISG responses in iF17-infected cells (Supplementary Fig. 5). Notably, subtle differences in viral DNA replication between each virus also correlated with the extent of mitochondrial aggregation around the Golgi (Supplementary Fig. 1b) and mtDNA release (Fig. 2b), suggesting that mitochondrial hyperfusion and mtDNA release is driven by the level of poxvirus replication in the cell.

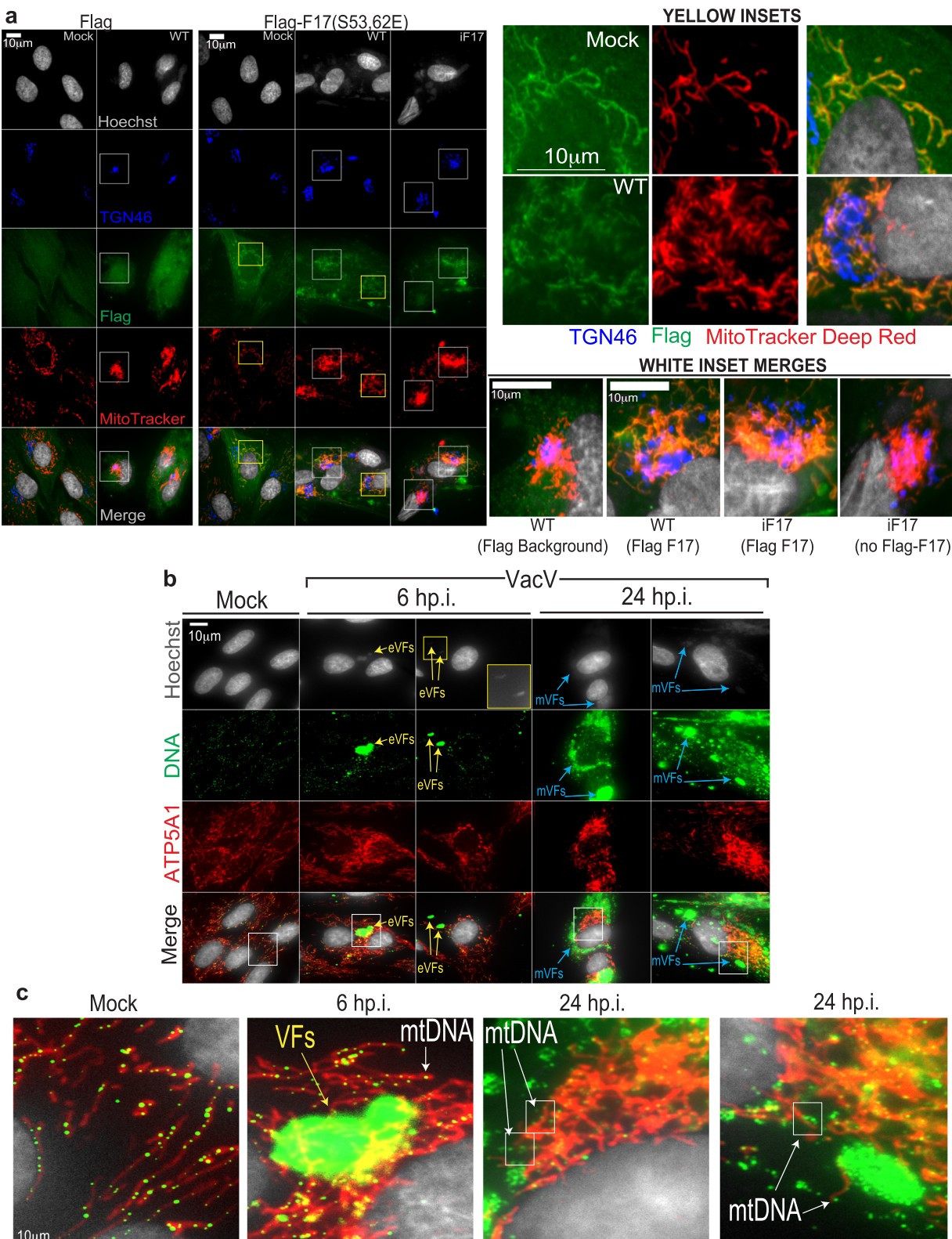

Western blot analysis further revealed that although VacV did not induce apoptosis or affect the levels of several core mitochondrial proteins examined, levels of the mitophagy factor, PINK1 together with phosphorylation at Ser[616] in the fission factor, DRP-1 were decreased upon infection (Fig. 2c). This would result in impaired mitochondrial fission and clearance[32]. By contrast, decreases in phosphorylation of Mitofusin 2 (MFN2) at Ser[422], which negatively regulates MFN2's fusion activity, were also observed. Combined, these changes suggest that

the balance of fission versus fusion activity is skewed towards fusion, which aligns with the accumulation of hyperfused mitochondria that is observed during VacV infection. Finally, Western blot analysis also demonstrated that the kinetics of ISG induction in response to iF17 infection correlated with the progression of virus replication into later stages (Fig. 2d and Supplementary Fig. 6) and with the kinetics of mtDNA release (Fig. 2b). Moreover, ISG responses to iF17 infection were suppressed by treatment of NHDFs with a small molecule

**Fig. 1 | Phosphorylated F17 localizes to mitochondria that leak mtDNA.** Cells in (a–c) were infected with the indicated viruses at MOI 5. **a** Flag versus Flag-F17-S53/62E expressing NHDFs infected with WT or iF17 virus were pulsed with MitoTracker Deep Red (demonstrating sustained MMP in infected cells), fixed at 24 h.p.i. and stained as indicated. Yellow Insets show Flag-F17 localization to mitochondria in either uninfected (mock) or infected (WT) cells. White Inset merges show background Flag peptide staining versus specific Flag-F17 staining at mitochondria in WT- or iF17-infected cells. For iF17-infected cells, a neighboring cell not expressing Flag-F17 illustrates mitochondrial aggregation does not require F17 expression.

**b**, **c** Examples of mitochondrial localization and anti-DNA antibody staining in mock versus VacV-infected NHDFs at 6 h.p.i or 24 h.p.i. Yellow arrows indicate small, early VFs (eVFs) that are readily detected by anti-DNA antibody staining but which are sometimes still so small that they require longer Hoechst exposures to visualize (middle top inset). Blue arrows indicate large, mature VFs (mVFs) evident by 24 h.p.i. Inset regions zoomed in c. illustrate mtDNA nucleoids inside mitochondria in mock infected cells and cells infected for 6 h. By 24 h.p.i., mtDNA nucleoids are more apparent outside mitochondria. **a–c** Representative of phenotypes in >90% of cells and observed in 3 or more independent experiments.

inhibitor of cGAS, even when added as late at 6 h.p.i. (Fig. 3a, b). Similarly, infection of WT or cGAS knockout THP1 monocytes that express an ISG54 promoter-driven Lucia Luciferase reporter demonstrated that responses to iF17 infection occurred at later times of infection and in a cGAS-dependent manner (Fig. 3c–e). Overall, these data demonstrated that mtDNA is released as VacV replication progresses and correlates with cGAS-dependent induction of ISG expression in the absence of F17.

To test if mtDNA activates these responses we first treated NHDFs with dideoxycytidine (ddC)[33]. ddC selectively inhibits mitochondrial but not nuclear DNA polymerase and is a widely used approach to deplete mtDNA and determine its contribution to immune activation[23]. ddC treatment efficiently reduced mtDNA levels in NHDFs prior to infection (Fig. 4a) and potently reduced its release into the cytosol during infection by iF17 (Fig. 4b). Moreover, ddC treatment resulted in a dose-dependent reduction in ISG expression in iF17 infected cells, without impacting viral protein production (Fig. 4c, d). ddC treatment also reduced ISG production and IRF reporter activation upon iF17 infection in THP1 monocytes (Fig. 4e, f), suggesting that mtDNA triggers these host responses. We also observed robust suppression of ISG responses in cells treated with Ethidium Bromide (EtBr), another commonly used approach to deplete mtDNA (Supplementary Fig. 7a–c). However, we interpret the magnitude of these effects with caution as unlike the specificity of ddC, EtBr indiscriminately intercalates into DNA and may therefore impact the quantity and/or quality of other DNA species that may also contribute to cGAS activation during infection.

Despite the greater specificity ddC, a caveat to any mtDNA depletion strategy is that this also reduces the expression of mitochondrially-encoded electron transport chain (ETC) subunits, such as cytochrome b (mtCytB) (Fig. 4c). This in turn reduces mitochondrial respiration or oxygen consumption rate (OCR) (Fig. 4g) and impairs the production of reactive oxygen species (ROS) and NLRP3 inflammasome activation[34,35]. This makes it difficult to discern with certainty the extent to which mtDNA depletion might affect cGAS sensing as opposed to impairing apoptotic and inflammasome-driven responses, which as discussed later are common during infection with many RNA viruses. However, VacV encodes an array of early proteins that limit the impact of infection on many aspects of mitochondrial activity, sustaining the Tricarboxylic Acid (TCA) cycle, MMP, ETC activity and ATP production while inhibiting inflammasomes, autophagy, necroptosis and apoptosis[36–48] (reviewed in[4,15]). In line with this, we have shown previously that VacV blocks autophagy even in the absence of F17[6] and here we show that mitochondria continue to stain brightly with the MMP-sensitive dye, MitoTracker Deep Red (Fig. 1a and Supplementary Figs. 1a, 3, 4a). Moreover, in NHDFs infected with either WT, iF17 or iF17R viruses, respiration was relatively unaffected (Fig. 5a) while ROS levels were reduced (Fig. 5b), rather than elevated as is associated with inflammatory responses to other viruses. To further test if ROS regulated responses to iF17 infection, we treated cells with the Complex III inhibitor Antimycin or the ATP synthase inhibitor Oligomycin, which decrease respiration and mitochondrial ATP production, and increase ROS[34,35]. In addition, we independently treated NHDFs with the Complex I inhibitor Piericidin to block mitochondria-based inflammasome activation[35]. While all of these inhibitors were

active in OCR assays, none of them had a significant effect on ISG responses to iF17 infection (Fig. 5c, d). Moreover, we found no evidence of increased apoptosis in NHDFs infected with WT, iF17 or iF17R viruses (Fig. 2c). Combined, these data demonstrated that in line with prior studies, VacV does not cause extensive mitochondrial damage or inflammasome activation. Instead, our findings reveal that VacV infection causes a more limited subset of mitochondrial responses that involve mtDNA release to activate antiviral responses in the absence of F17.

## Mitochondrial hyperfusion coordinates mtDNA release and increased glycolysis to fuel antiviral responses in the absence of F17

Recent studies have shown that some viruses cause mitochondrial hyperfusion that leads to controlled, sublethal release of mtDNA as a means to activate innate responses to infection[22,31]. Moreover, this process can be suppressed by depleting the fusion factor, Mitofusin 1 (MFN1). Similar to these reports, release of mtDNA during either WT or iF17 infection was suppressed by MFN1 depletion (Fig. 6a, b and Supplementary Fig. 8a). Furthermore, induction of ISGs in response iF17 infection was significantly reduced by depletion of MFN1 (Fig. 6a, c). However, unlike depletion of mtDNA, modulation of hyperfusion through MFN1 depletion did not affect mtCtyB expression (Fig. 6a) or mitochondrial respiration (Fig. 6d). This further supported the idea that mtDNA itself, and not effects on the ETC, underlie responses to iF17 infection.

During more extensive mitochondrial damage, mtDNA can also be released through apoptotic BAX/BAK macropores or Voltage Dependent Anion Channels (VDACs) in combination with reduced MMP[49–52]. In testing whether these alternative mechanisms of release also functioned during poxvirus infection, we found that ISG responses to iF17 were insensitive to treatment with Cyclosporin A (Fig. 7a). This agrees with several reports that VacV proteins block apoptotic pores and sustain MMP[15,37,39]. We also treated NHDFs with the VDAC oligomerization inhibitor, VBIT-4 that is reported to inhibit mtDNA release under pathological conditions such as lupus or infection by some RNA viruses[27,28,52]. However, not only did VBIT-4 not affect mtDNA release but it unexpectedly increased ISG responses to iF17 infection (Fig. 7b). Notably, VBIT-4 is not a specific inhibitor of mtDNA release but instead, it broadly reduces the impacts of apoptosis by stabilizing Hexokinase association with VDAC's, which sustains MMP and reduces ROS production but also drives the cell to switch to aerobic glycolysis[53]. This finding provides additional independent lines of evidence against roles for ROS in ISG responses to iF17 infection and also raised the possibility of a role for increased glycolysis.

Testing this further, we found that infection with either WT or iF17R viruses did not significantly affect extracellular acidification rates (ECAR) as a readout of glycolysis (Fig. 7c). However, infection with iF17 resulted in a significant increase in ECAR. This was independently confirmed by measuring intracellular lactate, the end product of the glycolysis pathway (Fig. 7d). Furthermore, treating NHDFs with the mTOR inhibitor, PP242[54] demonstrated that mTOR was required for the increase in both intracellular and extracellular lactate that is produced in response to infection with iF17 (Fig. 7e). This suggests that F17 functions to block a cellular increase in glycolysis that is controlled by

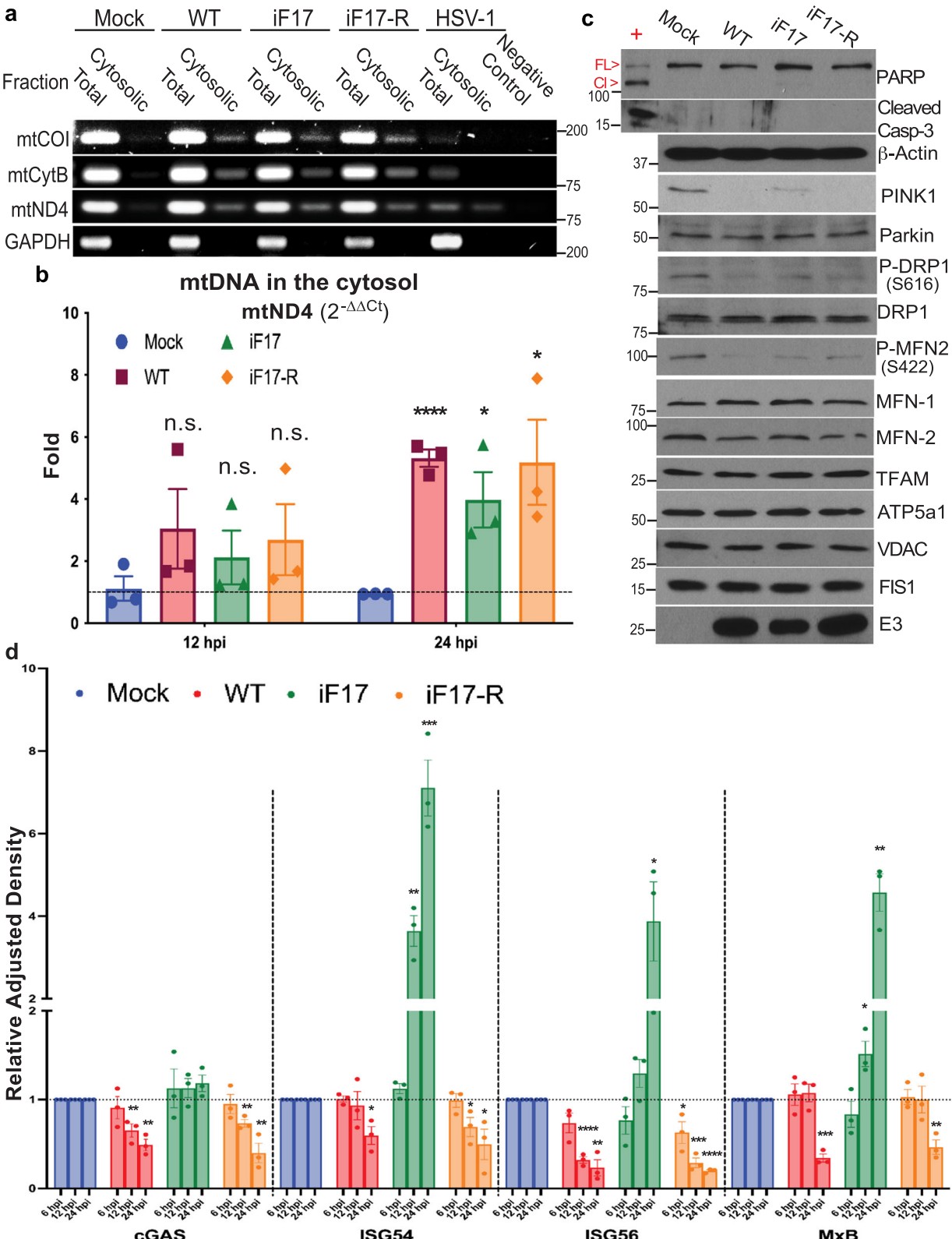

mTOR, a process known to play a key role in supporting innate immune responses[55,56]. Further testing the importance of glycolysis, we found that ISG production was significantly reduced by treatment of cells with either PFK15 to inhibit 6-phosphofructo-2-kinase (PFK2B3), a rate-limiting glycolytic enzyme, or 3′-bromopyruvate, a broad inhibitor of glycolysis (Fig. 8a, b). Inhibitors of glycolysis did not inhibit STING phosphorylation, suggesting that glycolysis independently supported innate responses by increasing the cells biosynthetic capacity. To

independently test these inhibitor-based findings, we also adapted NHDF cells to glucose-free medium that was supplemented with galactose. This reduces glycolytic flux while continuing to support mitochondrial functions for cellular energy demands[57]. Confirming the efficacy of adaptation, NHDFs cultured in glucose-containing medium were relatively insensitive to treatment with Piericidin, which inhibits the ETC, while extensive cell death was observed in Piericidin-treated NHDFs adapted to galactose due to their dependence on

**Fig. 2 | mtDNA leakage and ISG activation in the absence of F17 are later-stage events. a** PCR analysis of fractionated lysates shows increased mtDNA in the cytosol of NHDFs infected with WT, iF17 or iF17R viruses, and degradation of mtDNA by HSV-1. Molecular Weight markers are shown in bp. **b** RT-qPCR measurement of mtND4 levels in the cell cytosol of NHDFs infected with the indicated viruses at MOI 5 for either 12 h or 24 h, presented as fold-change over mock using the $2^{-\Delta\Delta Ct}$ method. n = 3 per group, multiple independent two-sided t-tests with the corresponding Mock for each time-point. Data are presented as mean values ± SEM. Note that although not statistically significant, cytosolic mtDNA levels are trending upwards in infected cells at 12 h.p.i. **c** NHDFs were infected at MOI 5 for 24 h. Levels and phosphorylation of mitochondrial proteins and apoptosis factors were assessed using the indicated antibodies. Cells treated with cycloheximide (CHX) served as a positive control for detection of caspase 3 (Casp-3) and PARP cleavage (FL=Full Length, Cl=Cleaved). Molecular Weight markers are shown in kDa. **d** Quantification of Western blots (see Supplementary Fig. 6 for examples) of cGAS and ISG levels in NHDFs infected with the indicated viruses at MOI 5 for 6 h, 12 h, or 24 h. n = 3 per group, multiple independent two-sided t-tests with the corresponding Mock for each protein. Data are presented as mean values ± SEM. To avoid clutter, only significant differences are indicated. **a–d** Representative of 3 or more biological replicates. For panels (**b**, **d**); n.s. = no significance, *p ≤ 0.05, **p ≤ 0.01, ***p ≤ 0.001, ****p ≤ 0.0001. Source data are provided as a Source Data file.

mitochondrial respiration (Fig. 8c). Upon infecting these cells, the ISG response to iF17 infection was significantly reduced in NHDFs that were adapted to galactose (Fig. 8d, e). Combined, these data demonstrated that cells increase glycolysis in an mTOR-dependent manner to support responses to poxvirus infection in the absence of F17.

Finally, to test potential connections between mtDNA release and glycolysis, we examined ECAR in ddC treated or MFN1-depleted cells. In uninfected cells, ddC treatment increased glycolysis (Fig. 9a), which would be expected due to decreased mitochondrial respiration as a result of reduced expression of ETC components (Fig. 4c, g). In iF17-infected cells, ddC treatment had no effect (Fig. 9c), in line with the fact that glycolysis was already elevated during infection. Given that glycolysis is active in iF17-infected cells yet ddC reduces ISG responses (Fig. 4), this suggests that mtDNA itself is required to activate cGAS and initiate ISG expression, irrespective of the levels of glycolysis. This also supports the broader notion that mtDNA and glycolysis independently control distinct aspects of the overall ISG response. Testing whether mitochondrial hyperfusion influenced glycolysis, we found that MFN1 depletion had a small but statistically insignificant effect on ECAR activity in uninfected cells (Supplementary Fig. 8b). Similarly, no significant impact on ECAR was detected in cells infected with WT virus where, unlike uninfected cells, extensive mitochondrial hyperfusion occurs (Fig. 9b). By contrast, MFN1 depletion resulted in a significant reduction in glycolysis in iF17-infected cells (Fig. 9b). This suggests that infection-induced changes in mitochondrial fusion act as a trigger for the cell to increase glycolysis, which is detectable in the absence of its inhibitor, F17. Combined, our data suggests that mitochondrial fusion coordinates two independent processes of mtDNA release to activate cGAS and an increase in glycolysis to support the biosynthesis of ISGs (Fig. 9c).

## Discussion

Since its discovery, cGAS has become recognized as a primary sensor of cytosolic DNA that arises in a variety of disease states[3]. Moreover, there is growing evidence that beyond the foreign DNA that is produced by several pathogens themselves, a number of viruses can also cause the release of host genomic and mtDNA into the cytosol to activate cGAS[16–29]. This offers an explanation for why some RNA viruses activate cGAS-mediated antiviral responses, yet it remains a complex question to directly address in several cases. Reported requirements for cGAS in limiting RNA virus infection may be indirect in at least some cases due, for example, to reduced basal ISG expression and increased permissiveness of knockout cells prior to infection[29]. Several of the RNA viruses reported to induce mtDNA leakage also induce and exploit autophagy and apoptosis, where mtDNA release, binding to cGAS and caspase-mediated cleavage or autophagic turnover of cGAS to limit inflammation are natural processes[49–51,58–60]. As a result, it can be difficult to tell if these events are driven by infection to counter a specific cGAS-mediated antiviral response or occur as part of infection-induced apoptotic processes. Most importantly, in many cases evidence that mtDNA itself plays a role in stimulating innate responses relies on the use of ddC or similar mtDNA depletion approaches. However, as we highlighted earlier, mtDNA depletion also affects the expression of ETC components that influence respiration, ROS and inflammasome activity. Given that inflammasomes play a major role in responding to many RNA virus infections, it becomes difficult to disentangle the effects of mtDNA depletion on inflammosome- versus cGAS-mediated antiviral responses. This complex interplay is highlighted by reports that find circular relationships between ROS production and mtDNA release[16,21,24,27]. As a result, while it is highly likely that mtDNA release and its sensing by cGAS contributes to antiviral responses to some RNA viruses, the underlying complexity makes it difficult to determine their specific contributions without the use of additional approaches that independently restore ROS production[35]. By contrast, poxviruses encode a multitude of inhibitors of these confounding processes[36–48] (reviewed in[4,15]) and, as we elaborate upon here, the effects of infection on mitochondrial functionality are far more limited. Interestingly, mitochondrial aggregation near the nucleus has been noted during infection with several different poxviruses but the significance of this is unknown[31,41,43,61–64]. Our findings reveal that this hyperfusion of mitochondria is a cellular response to infection that mediates mtDNA release and increases in glycolysis that support ISG production. This occurs regardless of F17's presence or absence and a similar mtDNA release process occurs during HSV-1 or MeV infection[22,23,31], further highlighting how this is likely to be a fundamental cellular response to several virus families.

mtDNA release would further add to the pressure that already exists for cytoplasmic DNA viruses such as VacV to counteract host sensors such as cGAS. While poxviruses encode a wide range of immunomodulatory proteins, it is only relatively recently that specific cGAS antagonists have begun to be identified[4]. These include B2 or "poxin", which is expressed early in infection and functions as a cGAMP nuclease[65]. Deletion of B2 does not result in IFN responses in the primary infected cell but significantly reduces viral load and pathogenesis in animal models[65,66]. This suggests that B2 primarily functions to limit the release of cGAMP from infected cells that would otherwise activate antiviral responses in neighboring cells. B2 is also not encoded by a number of poxviruses including MVA and VarV. The absence of B2 in virulent poxviruses such as VarV suggests the existence of additional cGAS antagonists[65]. Indeed, targeting critical innate response pathways at multiple levels is a common strategy amongst poxviruses[4]. In line with this, F17 is conserved across *Orthopoxviruses* and is expressed at later stages of infection, concurrent with the onset of extensive viral DNA replication and as we show here, virus-induced mtDNA release. In the case of F17 mutants, ISG induction is readily detected in the primary infected cell but as we have shown previously, other viral antagonists limit broader responses such as cytokine release[5]. It is also notable that B2 is not encoded by MVA, a highly attenuated strain of VacV that cannot counteract cGAS-mediated responses[67]. In most cell types, MVA also fails to express late genes, which include F17. Independent studies have shown that rescuing late gene expression restores MVA's ability to counteract host antiviral responses, which further suggests that late proteins such as F17 act as key antagonists of host responses during primary infection[68]. Combined, these findings suggest that B2 and F17 function at different times and in different ways to

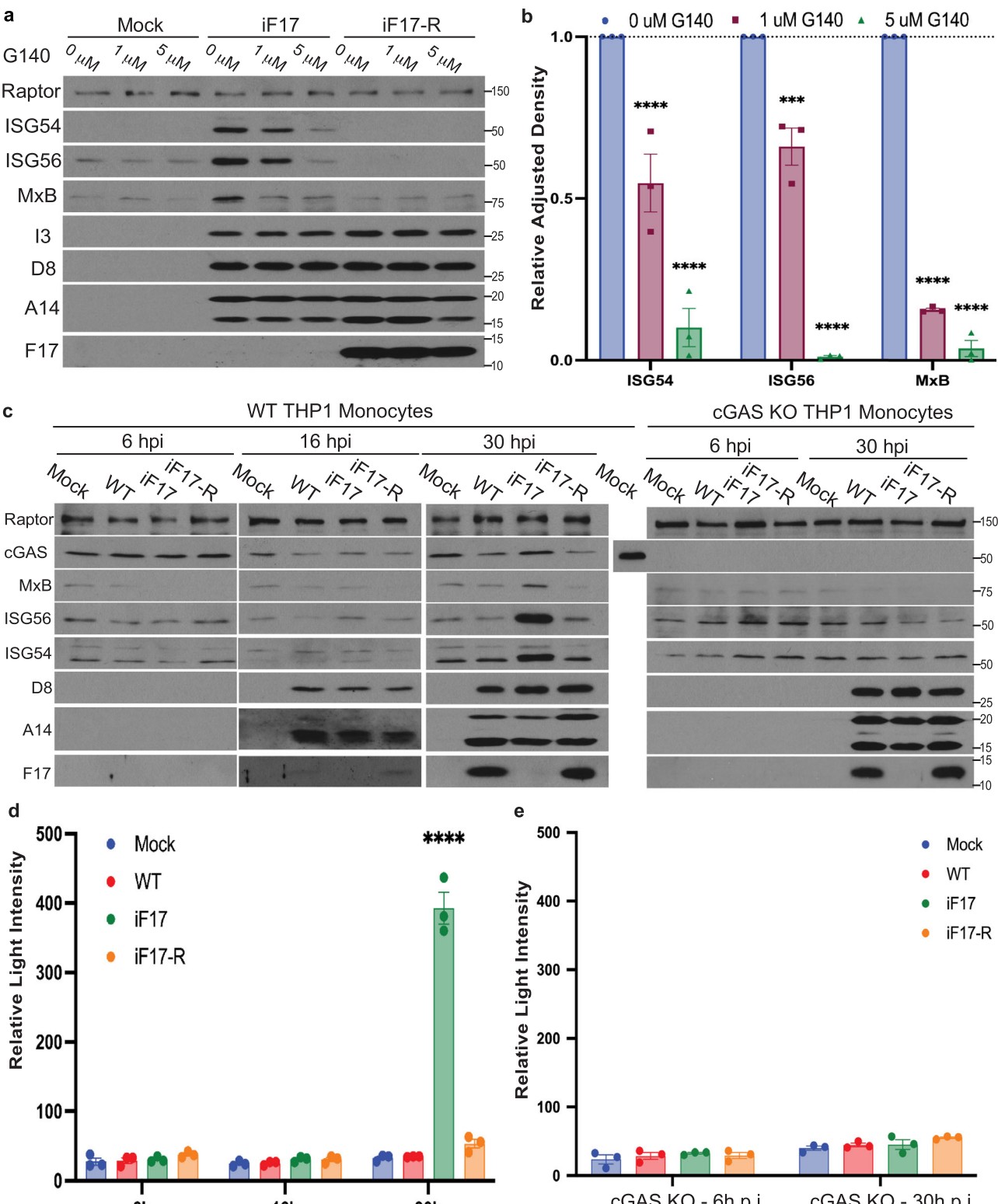

counteract different functions of cGAS activation over the course of infection.

Interestingly, while this manuscript was under review another poxvirus protein, named E5, was reported to be required to promote cGAS degradation very early in infection[69]. At first, this finding seems at odds with several prior studies and current thinking. First, E5 is a poorly characterized DNA binding protein that is expressed early in infection by several poxviruses, notably including MVA. The reported rapid and extensive degradation of cGAS seems counterintuitive to

MVA's reported deficiencies in blocking cGAS-mediated responses, particularly at later stages of infection[67,68,70]. Second, such rapid cGAS degradation has not been reported previously and by contrast, we only observe partial destabilization of cGAS at later stages of infection by WT VacV. However, a likely explanation for these apparent discrepancies lies in the fact that Yang et al. tested the role of E5 during infection in two contexts; MVA infections or WT VacV infection of primary murine Bone Marrow-derived Dendritic Cells (BMDCs)[69]. While MVA infection is abortive in many cell types, VacV is also thought

**Fig. 3 | cGAS is required for ISG responses to iF17 infection in NHDFs and THP1 monocytes. a, b** NHDFs were infected with the indicated viruses at MOI 5 and then treated at 6 h.p.i. with DMSO solvent control or the indicated concentrations of the cGAS inhibitor, G140. **a** Cells were lysed at 24 h.p.i. and analyzed by Western blotting using the indicated antibodies. Molecular Weight markers are shown in kDa. **b** Quantification of ISG levels in NHDFs described in a. n = 3 per group, 2-way ANOVA between concentrations followed by Sidak's multiple comparisons tests. Data are presented as mean values ± SEM. **c–e** WT or cGAS knockout (KO) THP1 monocytes stably expressing an ISG54 promoter-driven Lucia Luciferase reporter were infected at MOI 5 for the indicated times. **c** Cells were lysed and analyzed by

Western blotting using the indicated antibodies. As samples had to be run on separate gels, a WT THP1 cell lysate was run as a positive control for cGAS detection and to demonstrate the absence of cGAS in Knockout (KO) cells. Molecular Weight markers are shown in kDa. **d, e** Luciferase activity was measured in WT or cGAS KO THP1 monocytes infected as described in (**c**). n = 3 per group, 2-way ANOVA between time-points followed by Sidak's multiple comparisons tests. Data are presented as mean values ± SEM. For panels (**b, d, e**); ns = no significance, ***p ≤ 0.001, ****p ≤ 0.0001. All data are derived from or representative of 3 or more independent experiments. Source data are provided as a Source Data file.

to undergo abortive infections that induce apoptotic responses in certain primary immune cell types including Dendritic cells[71,72]. As such, while deletion of the gene encoding E5 represents a potentially valuable means to increase immune responses and vaccine efficacy, it remains to be determined whether E5 has similar functions during productive infection of other cell types.

Finally, our findings highlight the multifunctional nature of F17 that likely underlies its importance in counteracting antiviral responses at later stages of infection. While we focus here on the unexpected contribution of mtDNA towards sensing of a cytosolic DNA virus, it is notable that although inhibitors or genetic knockdown approaches are never 100% effective, responses were frequently dampened but not eliminated by reducing mtDNA levels or release. One exception was in EtBr-treated cells, which suggests that viral DNA or even leakage of nuclear DNA also contribute to cGAS activation. However, by destabilizing cGAS and impairing glycolysis, F17 disables key processes that are important for responding to diverse cytosolic DNA stimuli, including mtDNA. In terms of viral evasion strategies, while HSV-1 degrades mtDNA[22] we find that neither WT nor iF17 viruses block the release or promote the degradation of mtDNA. Instead, the viral F17 protein counteracts subsequent mitochondrially orchestrated responses by dysregulating mTOR to both destabilize cGAS[5] and as shown here, suppress glycolysis. Interestingly, unlike many other viruses that simultaneously activate all three key pathways to promote their replication[73], poxviruses fuel the TCA cycle through glutaminolysis and fatty acid oxidation but not glycolysis[42–44,47,48,74] (Fig. 9c), yet the reason for this is unknown. Our findings suggest that the F17 protein functions to actively prevent increases in glycolysis as this would otherwise fuel antiviral responses at later stages of replication. This may in turn explain why poxviruses evolved to utilize alternatives to glycolysis to sustain the TCA cycle. Finally, we also find that mtDNA release and increases in glycolysis independently contribute to the overall production of ISGs and that these processes are coordinated by mitochondrial fusion (Fig. 9c). Interestingly, MFN1 depletion only had significant impacts on glycolysis in infected cells in the absence of its inhibitor, F17, in agreement with the idea that infection-induced hyperfusion is an important driver of increases in glycolysis as part of the overall host response. The idea that multiple insults are required to drive a pathogenic response is common, to guard against inappropriate inflammation, and our findings align with studies of MeV where modulating hyperfusion outside the context of infection is not sufficient to cause ISG responses[31]. Indeed, the importance of F17 in blocking host responses to VacV infection undoubtedly lies in the fact that it does not selectively repress one process, such as mtDNA release, but broadly impairs host sensing and downstream response processes by localizing to mitochondria and dysregulating mTOR to simultaneously destabilize cGAS and suppress glycolysis.

## Methods
### Cell culture and viruses
Primary Normal Human Dermal Fibroblasts (NHDFs) isolated from male neonatal foreskin were purchased from Lonza (Cat# CC-2509) and cultured in complete Dulbecco's Modified Eagle's Medium (DMEM) supplemented with 5% Fetal Bovine Serum (FBS), 2 mM

L-Glutamine, and 1% penicillin-streptomycin. A retrovirus expression vector was used to create pools of low passage NHDFs expressing Flag, Flag-F17(S53,62E) and F17(S53,62A)-Flag[5,6]. WT and cGAS knockout THP1 Dual reporter cells were obtained from Invivogen (Cat# thpd-nfis, Cat# thpd-kocgas) and cultured in RPMI 1640 medium Fisher Scientific (Cat# 11875093) supplemented with 5% FBS, 2 mM L-Glutamine, and 1% penicillin-streptomycin. THP1 monocyte suspension cells were counted for viral infections using Trypan Blue, and 5 ×10$^5$ cells per well were used for 12-well plates. Female African green monkey BSC-40 cells were used to generate and titer Vaccinia Virus stocks[5,6]. All cell cultures were routinely screened and verified to be free of mycoplasma using DNA staining and imaging, as well as regular testing using commercial mycoplasma test kits.

Wildtype VacV (Western Reserve strain) was a gift from Dr. Stewart Shuman, Memorial Sloan Kettering. VacV iF17 (Western Reserve strain) was a gift from Dr. Bernard Moss, National Institutes of Health, and was propagated as described previously[5–7]. Briefly, stocks of iF17 virus were grown in the presence of 10 mM IPTG. Medium containing 10 mM IPTG was replaced every 24 h until approximately 90% cytopathic effect was observed in the BSC40 culture. The IPTG medium was then removed and fresh medium devoid of IPTG was used to wash the cells briefly for 5 min, this was then replaced with fresh medium for 2 h to wash out IPTG. The culture medium was again replaced and cells were harvested, freeze thawed three times, centrifuged to remove cell debris and stored as per WT or iF17R VacV. The iF17R revertant virus that no longer encodes the LacI repressor was plaque isolated from an iF17R pool, as described previously[6]. HSV-1 was grown and titered in the same manner as VacV stocks[5]. All infections were performed at Multiplicity of Infection (MOI) 5 for the indicated periods.

### Galactose adaptation of cells
Cells were cultured for 48 h in Glucose-free medium (ThermoFisher Scientific, Cat# A1443001) that had been filtered (ThermoFisher Scientific, Cat# 567-0010) and supplemented with 5% dialyzed FBS (PEAK SERUM, Cat# PS-FB1), 2 mM L-Glutamine, 1% penicillin-streptomycin, 1 mM Methyl-Pyruvate (Millipore Sigma, Cat# 371173) and 4.5 g/L D-(+)-Galactose (Millipore Sigma. Cat# G0750). Control NHDFs were cultured under the same conditions except using medium containing 4.5 g/L of glucose instead of D-(+)-Galactose. To ensure cells had become adapted to Galactose, cultures were treated with 1 μM Piericidin for 16 h and viable cell counts were determined using Trypan-Blue solution. For infections, cells were maintained in the indicated media throughout the course of infection.

### 5-ethynyl-2′-deoxyuridine labeling
Labeling of samples with 5-ethynyl-2′-deoxyuridine (EdU) was performed as described previously[75], with minor alterations to the protocol. In brief, mock infected or VacV infected cells were pulsed for 30 min with 10 μM Edu prior to Paraformaldehyde (PFA) fixation and cell permeabilization, as described below. Click-iT reactions to fluorescently label EdU incorporated into DNA were performed according to the manufacturer's protocol (Alexa Fluor-647 Click-iT Plus EdU Cell Proliferation Kit, ThermoFisher Scientific, Cat#C10640). Samples were then stained with antibody against the viral A14 protein together with

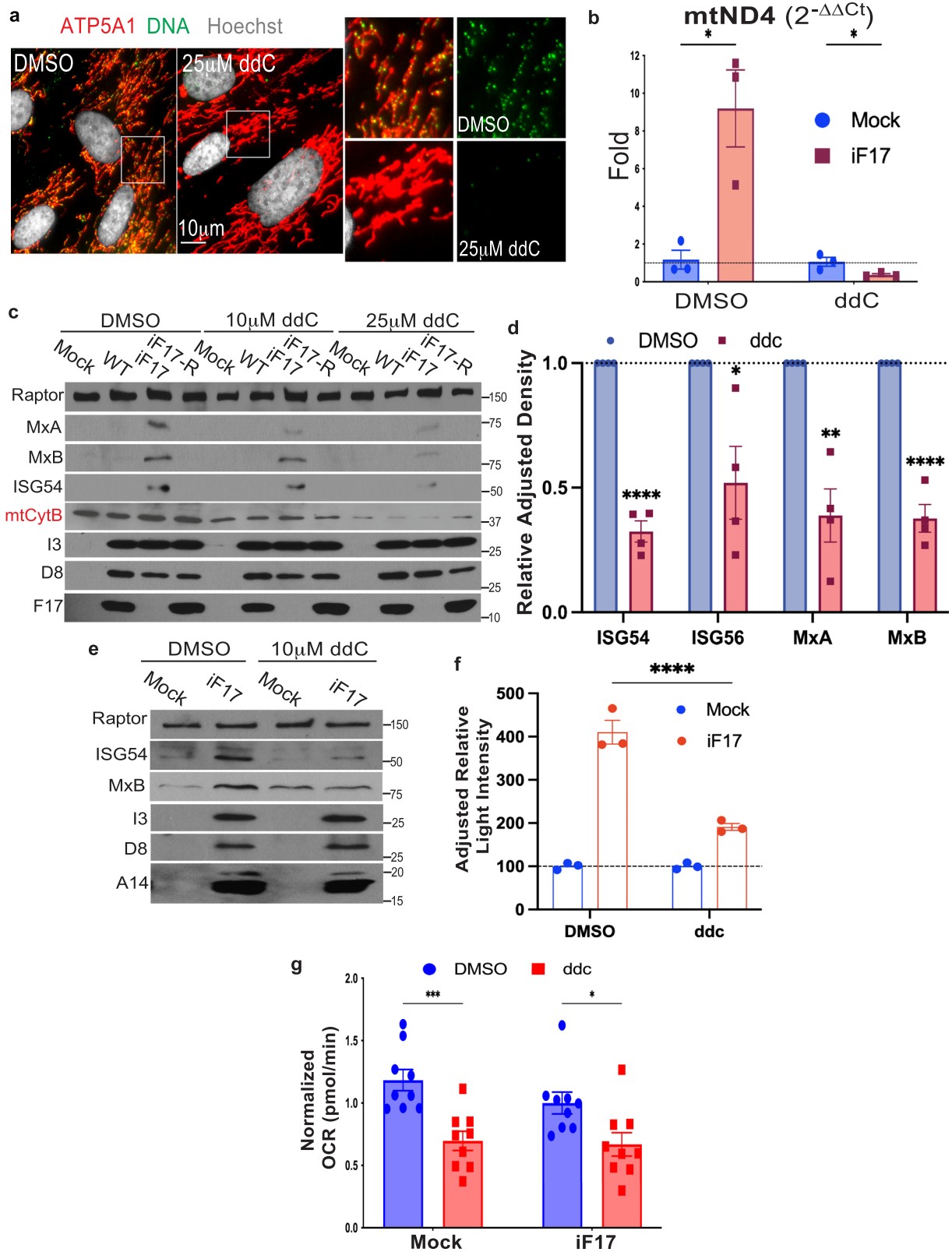

Hoechst. A14 and Hoechst were used as object identifiers for viral factories for analysis of EdU fluorescence using CellProfiler, as described below and previously[75].

**Immunofluorescence microscopy and imaging analysis**

Immunofluorescence (IF) was performed as described previously[5,75]. Briefly, cells were fixed for 20 min in PFA or for 7 min in Methanol,

depending on the antigen in question, and then permeabilized in 0.1% Triton-X100 in PBS for 30 min. Coverslips were then blocked in PBS containing 10%FBS and 0.25% Saponin for 1 h and then incubated with the indicated primary antibodies overnight at 4 °C at 1:100-200 dilution. After washing in PBS containing 0.025% Saponin, samples were incubated with the appropriate AlexaFluor-conjugated secondary antibodies for 1 h at room temperature. Samples were then washed and

**Fig. 4 | Depletion of mtDNA reduces ISG responses to iF17 infection. a, b** NHDFs were treated with DMSO control or 25 μM ddC for 4 days. **a** Fixed cells were stained for DNA to confirm loss of mtDNA prior to infection. **b** RT-qPCR analysis of cytosolic fractions shows reduced mtDNA in the cytosol of iF17 infected cells treated with ddC. Samples are normalized to their respective mock to measure relative release as ddC reduces mtDNA levels overall. n = 3 per group, independent two-tailed t-tests. Data are presented as mean values ± SEM. **c** ddC treatment has a dose-dependent effect on ISG responses to iF17 infection but also reduces mitochondrial cytochrome b (mtCytb) levels in NHDFs. Molecular Weight markers are shown in kDa. **d** quantification of ISG expression in iF17-infected NHDFs treated with DMSO control or 25 μM ddC. n = 4 per group, independent two-tailed t tests. Data are

presented as mean values ± SEM. **e, f** THP1 monocytes expressing an IRF reporter were treated with 10 μM ddC for 6 days prior to iF17 infection. **e** Western blot analysis of ISGs and viral proteins. Molecular Weight markers are shown in kDa. **f** Luciferase assays measuring IRF activity. n = 3 per group, 2-way ANOVA, Sidak's multiple comparison test. Data are presented as mean values ± SEM. **g** Oxygen consumption rate (OCR) is reduced in 25 μM ddC-treated NHDFs. n = 9 per group, independent two-tailed t tests. Data are presented as mean values ± SEM. **a–g** Representative of 3 or more biological replicates. For panels (**b, d, f, g**); *p ≤ 0.05, **p ≤ 0.01, ***p ≤ 0.001, ****p ≤ 0.0001. Source data are provided as a Source Data file.

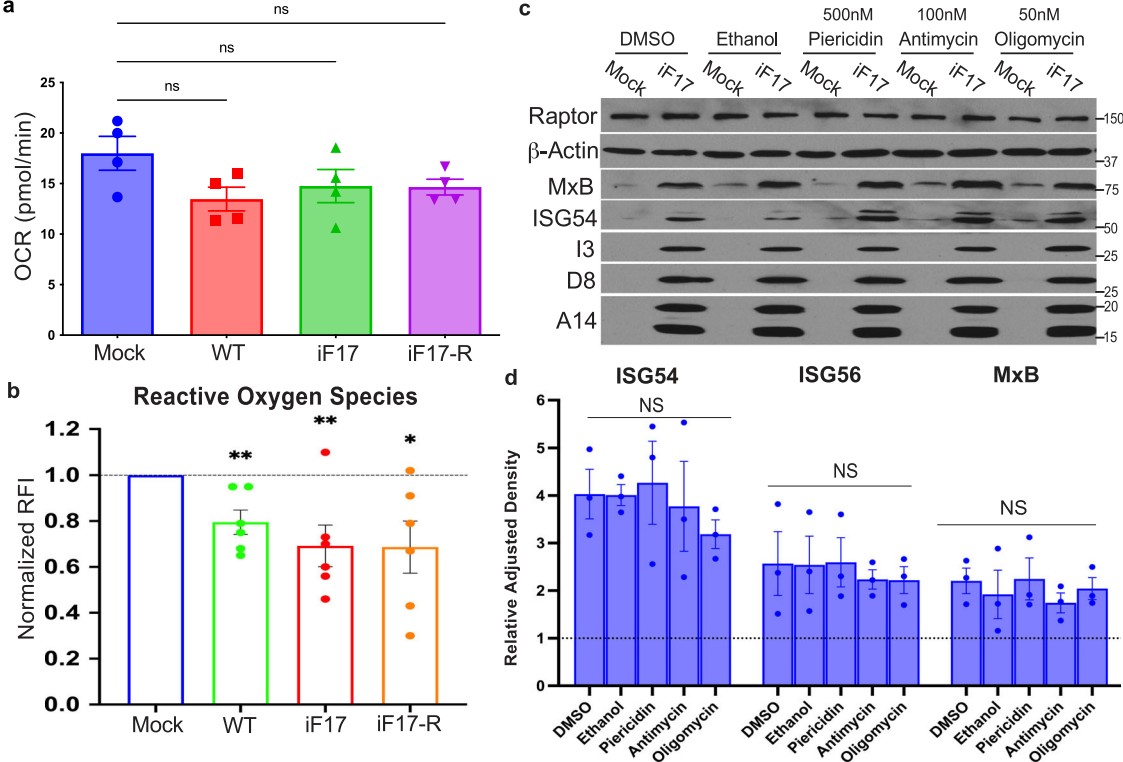

**Fig. 5 | The ETC and ROS do not control responses to iF17 infection. a, b** NHDFs were infected at MOI 5 for 24 h. **a** Oxygen consumption or (**b**), reactive Oxygen species were measured. n = 4 per group, independent two-tailed t tests. Data are presented as mean values ± SEM. **c, d** Cells were treated with solvent controls (DMSO or EtOH), 50 nM Piericidin, 100 nM Antimycin or 50 nM Oligomycin at the time of infection. **c** representative Western blots of ISG and viral protein levels.

Molecular Weight markers are shown in kDa. **d** Densitometry was used to quantify ISG levels, presented relative to uninfected controls under each condition. n = 3 per group, 2-way ANOVA, Dunnet's multiple comparison test. Data are presented as mean values ± SEM. For (**a, b, d**); ns = no significance, *p ≤ 0.05, **p ≤ 0.01. **a–d** Representative of 3 or more biological replicates. Source data are provided as a Source Data file.

stained with a 1:1500 dilution of 20 mM Hoechst 33342 Solution (ThermoFisher Scientific, Cat# 62249) prior to mounting on slides using FluroSave (Calbiochem: 345789). Details of primary and secondary antibodies used in this study are provided in the 'antibodies and reagents' section below. For MitoTracker staining, cells were incubated for 30 min with MitoTracker Deep Red (ThermoFisher Scientific, Cat# M22426) at a concentration of 100 nM in complete Phenol Red free DMEM, prior to fixation with 4% PFA in Phosphate Buffered Saline (PBS). Imaging was performed using a wide-field Leica DMI6000B-AFC microscope with 100X objective (HC PL APO 100x/1.44NA OIL), X-Cite XLED1 illumination, and an ORCA FLAH 4.0 cMOS camera. Metamorph Microscopy Automation and Image Analysis Software, Molecular Devices, (https://www.moleculardevices.com/products/cellular-imaging-systems/acquisition-and-analysis-software/metamorph-microscopy) was used for image acquisition and the multi-dimensional acquisition function was used to standardized settings for all images gathered in each independent experiment. Final

image analysis and processing was done using Metamorph and figures were compiled using the Fiji distribution of ImageJ[76] (https://fiji.sc/). For slide scan analysis, an automated stage was used to image the total area of each coverslip. Images were then entered into the CellProfiler Image Analysis Software program pipeline and resized to 256×256 pixels[75,77] (https://cellprofiler.org/releases). The Trans-Golgi-Network (TGN) was identified using TGN46 staining. Locations were marked based on the fluorescent intensity relative to the background. Cell profiler was then used to measure the fluorescence intensity generated by the aggregation of mitochondria around the TGN in a 10 μm radius. Mitochondria were identified using a primary antibody to a mitochondrial localized protein or MitoTracker Deep Red. The mean intensity measurements were used to analyze the data generated and for graphical representation with GraphPad Prism (version 9.0, GraphPad Software Inc, https://www.graphpad.com/scientific-software/prism/). All graphical representations in this study used GraphPad Prism.

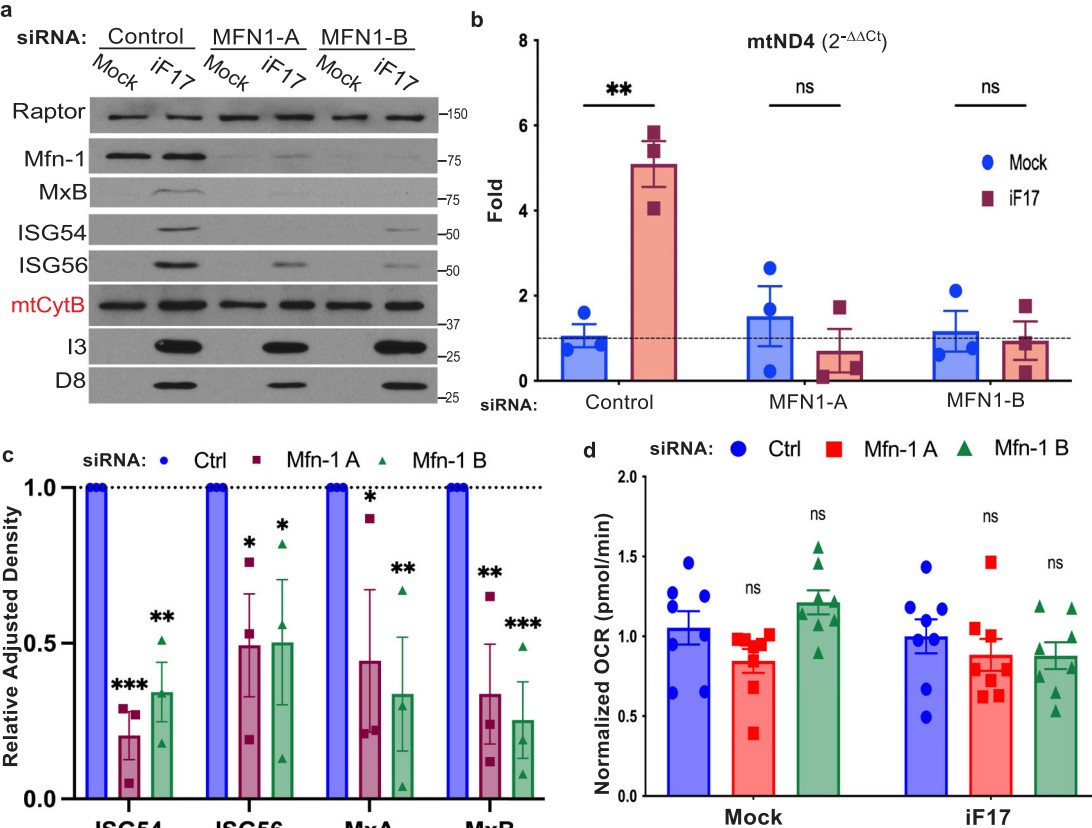

**Fig. 6 | Mitochondrial fusion regulates mtDNA release and responses to iF17 infection. a–d** NHDFs were treated with control or independent MFN1 siRNAs prior to infection. **a** Representative Western blot showing MFN1 depletion reduces ISG responses to iF17 infection but does not affect mtCytB levels. Molecular Weight markers are shown in kDa. **b** RT-qPCR analysis of fractionated samples showing relative levels of cytosolic mtDNA presented as fold change over control siRNA-treated mock infected cells, arbitrarily set to 1. n = 3 per group, unpaired two-tailed t-test. Data are presented as mean values ± SEM. **c** Densitometry was used to quantify ISG levels, presented relative to uninfected controls. n = 3 per group, 2-way ANOVA, Sidak's multiple comparison test. **d** MFN1 depletion does not affect oxygen consumption rates (OCR). n = 3, 2-way ANOVA, Sidak's multiple comparison test for control versus MFN1 siRNA treatments. Data are presented as mean values ± SEM. **a–d** Representative of 3 or more biological replicates. For (**b, c, d**); ns = no significance, *$p \leq 0.05$, **$p \leq 0.01$, ***$p \leq 0.001$. Source data are provided as a Source Data file.

## Western blotting

Sodium Dodecyl Sulfate Polyacrylamide Gel Electrophoresis (SDS-PAGE) and Western blotting were performed as previously described[5,6]. Briefly, to generate whole cell lysates, cultures were lysed in-well in Laemmli buffer. For fractionated samples, lysates were mixed with 2x Laemmli. Samples were boiled for 3 min, cooled and then run through an SDS-PAGE gel alongside molecular weight markers. Resolved samples were transferred to a Protran Nitrocellulose Membrane (Fisher Scientific, Cat# 45004006) which was then blocked by rocking in 5% non-fat milk solution in Tris buffered saline-Tween (TBS-T) for 1 h at room temperature (RT). Blocked membranes were then rinsed in TBS-T and incubated by rocking overnight at 4 °C with the appropriate primary antibody (see 'antibodies and reagents' below) diluted in TBS-T containing 3% bovine serum albumin. Membranes were then rinsed, washed three times in TBS-T and incubated for 1 h at RT with the appropriate HRP-conjugated secondary antibody (see 'antibodies and reagents' below). Membranes were then rinsed, washed again three times in TBS-T and protein bands were detected using Enhanced Chemiluminescence.

## Antibodies and reagents

Primary antibodies used in this study were obtained from the following sources and used for Western blotting at 1:1000 dilution unless otherwise stated:

Rabbit anti-Raptor, Cell Signaling Technology, Cat# 2280

Rabbit anti-cGAS, Cell Signaling Technology, Cat# 15102
Rabbit anti-cGAS, PROTEINTECH, Cat# 26416-1-AP
Rabbit anti-STING, Cell Signaling Technology, Cat# 13647 S
Rabbit anti-phospho-STING (S366), Cell Signaling Technology, Cat# 85735 S
Rabbit anti-MxA, Cell Signaling Technology, Cat# 37849
Rabbit anti-MxB, PROTEINTECH, Cat# 13278-1-AP
Rabbit anti-MxB, NOVUS BIOLOGICALS, Cat# NBP1-81018
Rabbit anti-ISG56, Cell Signaling Technology, Cat# 14769
Rabbit anti-ISG54, PROTEINTECH, Cat# 12604-1-AP
Sheep anti-TGN46, Bio-Rad, Cat# AHP500
Mouse anti-Flag M2, MilliporeSigma, Cat# F1804
Rabbit anti-Lamin A/C, PROTEINTECH, Cat# 10298-1-AP
Rabbit anti-FIS1, PROTEINTECH, Cat# 10956-1-AP
Rabbit anti-ATP5a1, PROTEINTECH, Cat# 14676-1-AP
Mouse anti-DNA, MilliporeSigma, Cat# CBL186
Rabbit anti-Pink1, Abcam, Cat# ab216144
Rabbit anti-PARK2/Parkin, PROTEINTECH, Cat# 14060-1-AP
Rabbit anti-Mitofusin-1, Cell Signaling Technology, Cat# 14739
Rabbit anti-Mitofusin-2, Cell Signaling Technology, Cat# 9482
Rabbit anti-phospho-Mitofusin-2 (S442), Millipore Sigma, Sigma Aldrich, Cat# ABC963
Rabbit anti-VDAC1/2, PROTEINTECH, Cat# 10866-1-AP
Rabbit anti-TFAM, PROTEINTECH, Cat# 22586-1-AP
Rabbit anti-CytB, PROTEINTECH, Cat# 55090-1-AP
Rabbit anti-DRP1, PROTEINTECH, Cat# 12957-1-AP

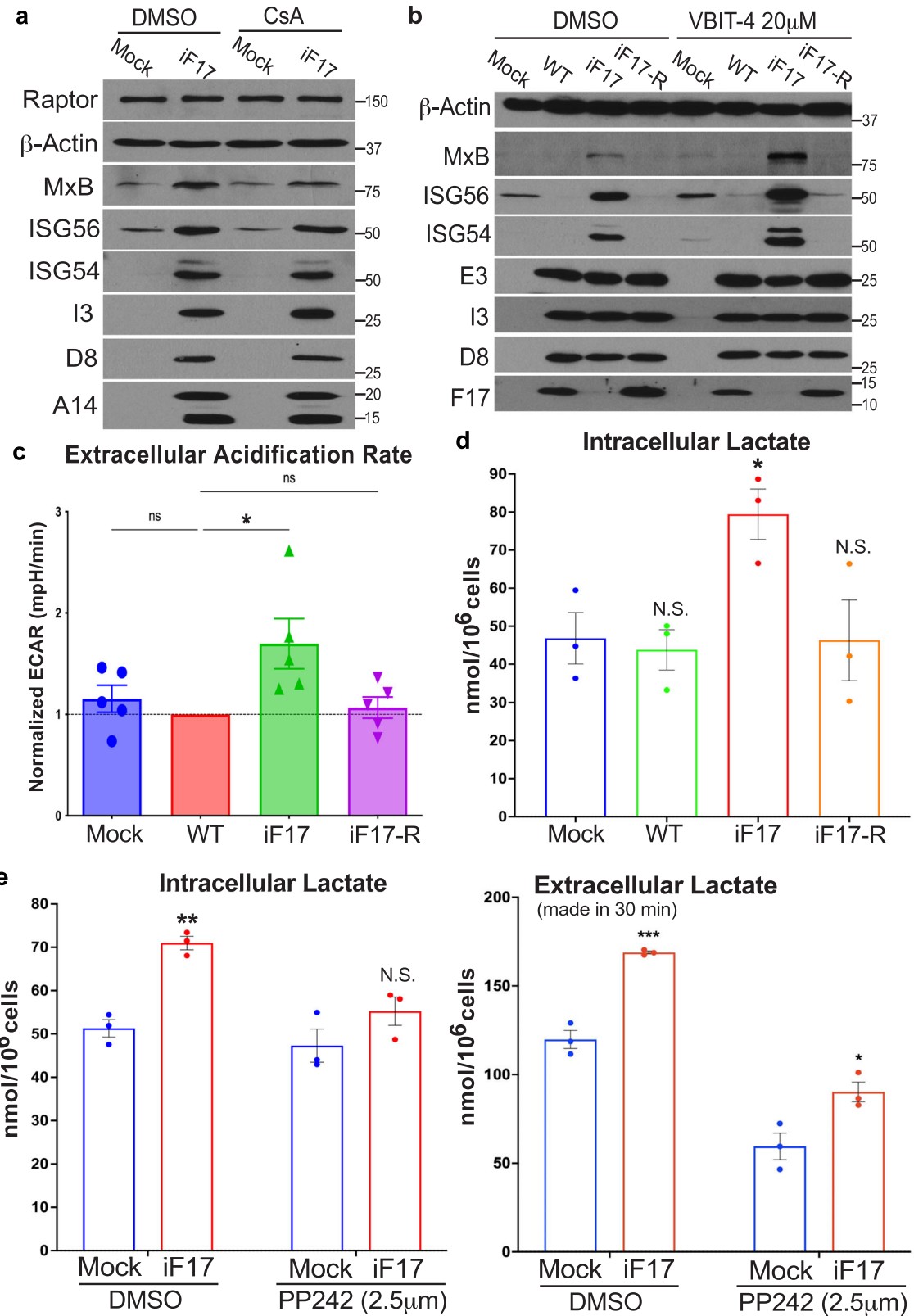

Rabbit anti-phospho-DRP1 (S616), Cell Signaling Technology, Cat# 3455

Rabbit anti-PARP, Cell Signaling Technology, Cat# 9542

Rabbit anti-Cleaved Caspase-3 (Asp175), Cell Signaling Technology, Cat # 9661

Mouse anti-β-Actin, Cell Signaling Technology, Cat# 3700

Rabbit anti-VacV F17 was raised against full length purified F17 protein by ABclonal, USA

Mouse anti-VacV I3 was a kind gift of Dr. David Evans

Mouse anti-VacV E3 was a kind gift of Dr. Jingxin Cao

Mouse anti-VacV D8 was a kind gift of Dr Paula Traktman

Mouse anti-VacV A14 was a kind gift of Dr. Yan Xiang

**Fig. 7 | Glycolysis is increased in response to infection in the absence of F17. a–e** NHDFs were infected at MOI 5 for 24 h. Cells were treated with DMSO or (**a**), 5 μM cyclosporin A (CsA) or (**b**), 20 μM VBIT-4 at the time of infection. Representative blots show effects of each inhibitor on ISGs and viral proteins. Molecular Weight markers are shown in kDa. **c** Extracellular acidification rate (ECAR) is increased in NHDFs infected with iF17. n = 5 per group, normalized to WT VacV-infected samples, independent two-tailed $t$ tests. Data are presented as mean values ± SEM. **d** Intracellular lactate in NHDFs at 24 h.p.i. n = 3, independent two-tailed $t$ tests with Mock. **e** Lactate production during responses to iF17 infection requires mTOR activity. DMSO or 2.5 μM of the mTOR inhibitor PP242 were added at 6 h.p.i. Intracellular (left) or released extracellular lactate in a 30 min period (right) was measured after medium change. n = 3 per group, independent two-tailed $t$ tests within treatments. Data are presented as mean values ± SEM. For (**c**, **d**, **e**); ns = no significance, *p ≤ 0.05, **p ≤ 0.01, ***p ≤ 0.001. **a–e** Representative of 3 or more biological replicates. Source data are provided as a Source Data file.

Secondary antibodies used in this study were obtained from the following sources and used at 1:3000 dilution;

Anti-Mouse IgG, HRP, Millipore Sigma, Cat# NA931V
Anti-Rabbit IgG, HRP, Millipore Sigma, Cat# NA934V
Donkey Anti-Mouse secondary Alexa 488, ThermoFisher Cat# A21202
Donkey Anti-Mouse secondary Alexa 555, ThermoFisher Cat# A31570
Donkey Anti-Mouse secondary Alexa 647, ThermoFisher Cat# A31571
Donkey Anti-Rabbit secondary Alexa 488, ThermoFisher Cat# A21206
Donkey Anti-Rabbit secondary Alexa 555, ThermoFisher Cat# A31572
Donkey Anti-Rabbit secondary Alexa 647, ThermoFisher Cat# A31573
Donkey Anti-Sheep secondary Alexa 488, ThermoFisher Cat# A11015
Donkey Anti-Sheep secondary Alexa 555, ThermoFisher Cat# A21436

Inhibitors used in this study were obtained from the following sources;

DMSO, Millipore Sigma, Cat# D2438
PP242, Millipore Sigma, Cat# 475988
2′,3′-Dideoxycytidine (ddc), Millipore Sigma, Cat# D5782
Zalcitabine (2′-3′-dideoxycytidine, ddC), Selleckchem, Cat# S1719
Ethanol 200 Proof Decon Labs, Fisher Scientific, Cat# 04-355-450
Piericidin, Cayman Chemical, Cat# 15379
Antimycin A, Millipore Sigma, Cat# A8674
Oligomycin, Millipore Sigma, Cat# 75351
Cyclosporin A, Millipsore Sigma, Cat# C1832
VBIT4, Selleckchem, Cat# S3544
Bromopyruvic acid/ 3-Bromo Pyruvate, Selleckchem, S5426
PFK15, Selleckchem, Cat# S7289
Cycloheximide, Research Products International, Cat# 81040
Human cGAS Inhibitor G140, Fisher Scientific, Cat# NC1924649
Ethidium Bromide, Fisher Scientific, Cat# BP1302-10

Details of inhibitor concentrations and periods of treatment are provided in figures and their related legends. Each stock solution was diluted in medium and 50 μl was added to cell cultures to reach the desired final concentrations. DMSO or Ethanol was used as the carrier for the inhibitors, and equal amounts of these carriers were used as control treatments. Complete media used for ddc or Ethidium Bromide treatment contained 500 μM Uridine (Millipore Sigma, Cat# U3750) and 1 mM pyruvate solution (Agilent, Cat# 103578-100). Optimal conditions for mtDNA depletion were determined to be 25 μM ddc for 4d in primary NHDFs or 10 μM ddc for 6d in THP1 cells, with the ddc being replenished every 48 h, and 1 μg/ml Ethidium Bromide for 2d in primary NHDFs. For EtBr experiments, EtBr was washed from cells at the time of infection.

### Cell lysis and fractionation
Cell fractionation for analysis by Western Blot, conventional Polymerase Chain Reaction (PCR) or quantitative PCR (qPCR) was carried out as described previously, with minor modifications[14,18,22]. Briefly, cells seeded in 6-well plates were washed in 1 ml PBS and then trypsinized for 2 min using 0.5 ml Trypsin. Following this, 3 ml complete DMEM was added and cells centrifuged at 300 × $g$ for 4 min. Cells were washed in PBS and again pelleted at 300 × $g$ for 4 min. Cell pellets were then resuspended in 300 μl Digitonin Buffer (50 mM NaCl, 50 mM Hepes pH7.4, 20 μg.ml$^{-1}$ Digitonin, R$_o$ H$_2$O) rocking for 10 min at 4 °C. Samples were then passed through a 26.5 gauge needle 10 times to complete cell lysis. A sample of the lysate was taken as a "total lysate" before the remaining lysate was centrifuged at 1000 × $g$ for 10 min to generate the nuclear pellet, followed by 20,000 × $g$ for 20 min for the Mitochondrial pellet, and the remaining supernatant was retained as the cytosolic fraction.

For analysis of proteins in fractions, the total, fractionated pellets and cytosolic supernatant were mixed with 50 μl Laemmli Buffer (62.5 mM Tris-HCl at pH 6.8, 2% SDS, 10% glycerol, 0.7 M β-mercaptoethanol, trace amount of bromophenol blue). Samples were then resolved by SDS-PAGE and analyzed by Western blotting, as described above. For analysis of DNA in fractions, DNA was extracted from total and cytosolic fractions as per the manufacturer's instructions (IBI Genomic DNA Mini-kit, Cat# IB47201). DNA concentrations were determined using a ThermoFisher Scientific NanoDrop 8000. PCR conditions were as follows; a standardized volume of DNA across all samples, 1x Q5 PCR Buffer NEB (Cat # B9027), 2U Q5 High-Fidelity DNA Polymerase NEB (Cat # M0491), 0.3 mM dNTPs NEB (Cat # N0447), Forward Primer (0.3 μM), Reverse Primer (0.3 μM), made up to 50 μl using RNase Free H$_2$O. PowerUp SYBR Green Master Mix (ThermoFisher Scientific, Cat# A25741) was used as per the manufacturer's guidelines. For conventional PCR, the following thermocycle parameters were used; 1 cycle of 94 °C for 30 s, 25–30 cycles of 98 °C for 10 s - 53 °C for 30 s - 72 °C for 30 s, and 1 cycle of 72 °C for 420 s, samples were then held at 4 °C. Thermocycling parameters for qPCR differed by gene and cell type. PCR products were resolved on a 1% agarose−TAE gel (Tris base, acetic acid, and EDTA buffer) at 110 V for 1 h using SYBR Safe (ThermoFisher Scientific, Cat # S33102) for visualization of the amplified regions under UV light. Cycle threshold (Ct) values obtained by qPCR were used to analyze mtDNA products from the total and cytosolic fractions of VacV infected samples. These were standardized to the uninfected samples using the $2^{-\Delta\Delta Ct}$ method and validated with the nuclear gene GAPDH, as described previously[18,22]. For PCR analysis of viral DNA levels, the following thermocycle parameters were used; 1 cycle of 95 °C for 120 s followed by 50 cycles of denaturing at 95 °C for 15 s and annealing/extension at 60 °C for 60 s, and samples were then held at 4 °C[78]. Primers used for PCR reactions were as follows, with reactions containing H$_2$O acting as negative controls:

VacV I3 Forward Primer: CAACAAATTAAACGGAGCCAT
VacV I3 Reverse Primer: CCTTGGCCAATTGTCTTTCTC
mtND4 Forward Primer: CCTCGTAGTAACAGCCATTC
mtND4 Reverse Primer: TTGAAGTCCTTGAGAGAGGA
mtCytB Forward Primer: CCATCCTCCATATATCCAAA
mtCytB Reverse Primer: CCAATGATGGTAAAAGGGTA
mtCOI Forward Primer: ATTCATCGGCGTAAATCTAA
mtCOI Reverse Primer: AGGCTTCTCAAATCATGAAA
GAPDH Forward Primers: GAGTCAACGGATTTGGTCGT
GAPDH Reverse Primers: TTGATTTTGGAGGGATCTCG

### RNA interference
Small interfering RNAs (siRNAs) were acquired from ThermoFisher Scientific: MFN1 Silencer Select Pre-designed siRNA A (Cat# 4392420,

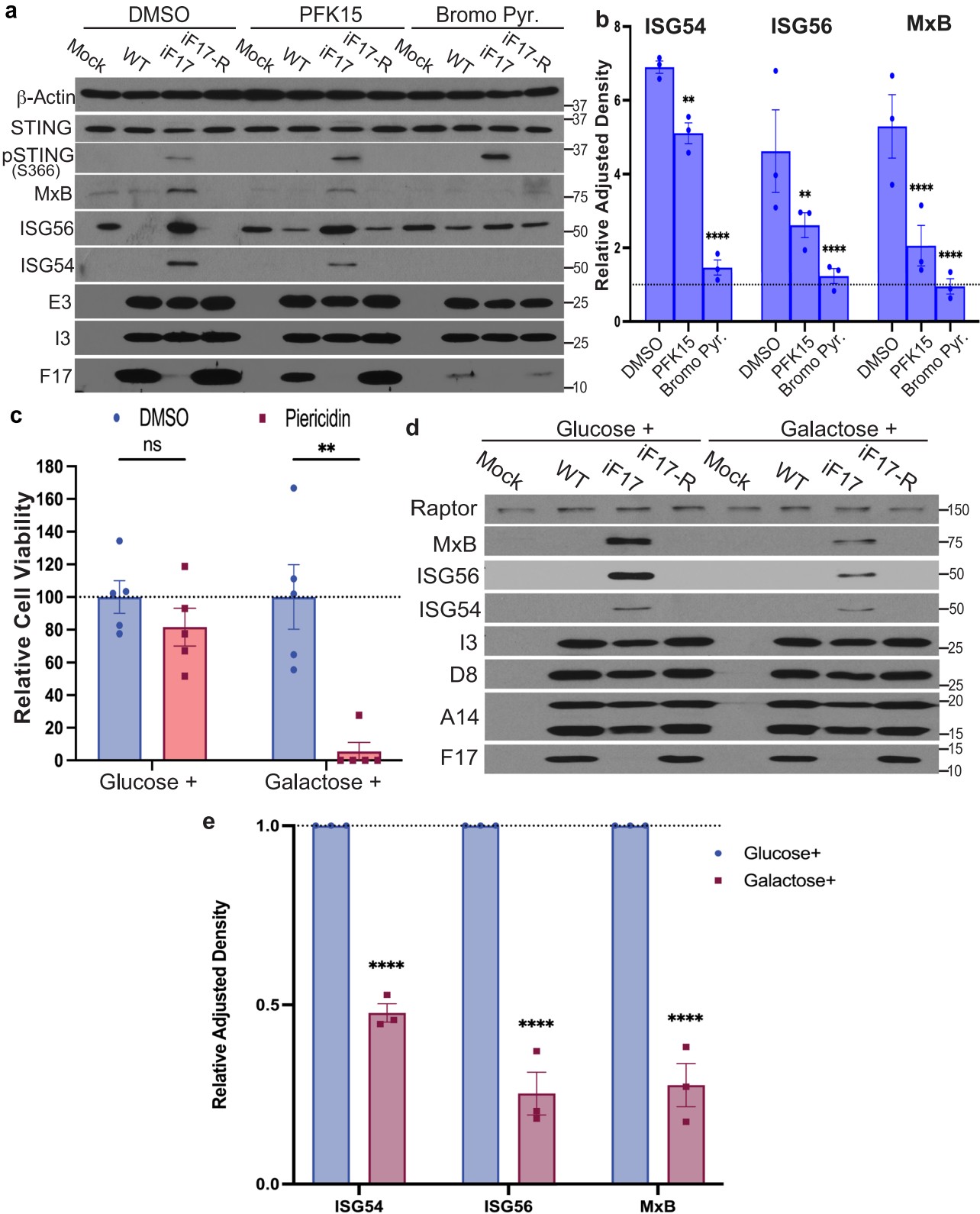

ID: s31219), MFN1 Silencer Select Pre-designed siRNA B, (Cat # 4392420, ID: s31220) and Silencer Negative Control siRNA (Cat#: AM4635). A 100 pmol.ml⁻¹ concentration of siRNAs were transfected into NHDFs that were approximately 60–70% confluent using Opti-MEM and Lipofectamine RNAiMAX (ThermoFisher Scientific, Cat #13778030) as per the manufacturer's guidelines. Following a transfection period of 4 h, complete DMEM was added and cultures were incubated for 72 h prior to analysis or infection, as detailed in the results section and figure legends.

## IRF luciferase reporter assay

Monocytic THP-1 dual reporter cells were used to determine activity from the IRF family of proteins by expression of Lucia luciferase and detection with Quanti-Luc InvivoGen (Cat# rep-qlc). This reporter is

**Fig. 8 | Glycolysis is required for ISG responses to iF17 infection. a–e** NHDFs were infected at MOI 5 for 24 h. **a, b** Cells were treated with 10 µM PFK15 or 200 µM 3-Bromopyruvate at 10 h.p.i. **a** Representative Western blots. Molecular Weight markers are shown in kDa. **b** Quantification of effects of inhibitors on ISG responses to iF17 infection. n = 3 per group, 2-way ANOVA, Tukey's multiple comparison within treatments. Data are presented as mean values ± SEM. **c–e** NHDFs were cultured in medium containing glucose or galactose to sustain or suppress glycolysis, respectively. **c** Cells were treated with 1 µM Piericidin to inhibit the ETC. Cell viability was measured using Trypan blue. n = 4 per group, multiple independent *t* tests with the corresponding solvent control for each treatment. Data are

presented as mean values ± SEM. **d, e** Glucose- or galactose-adapted NHDFs were infected with the indicated viruses at MOI 5 for 24 h. **d** Representative Western blots using the indicated antibodies. Molecular Weight markers are shown in kDa. **e** Densitometry was used to quantify ISG levels, presented relative to uninfected controls under each condition. n = 3 per group, 2-way ANOVA between treatments followed by Sidak's multiple comparisons tests. Data are presented as mean values ± SEM. For (**b, c, e**); ns = no significance, **p ≤ 0.01, ***p ≤ 0.001, ****p ≤ 0.0001. **a–e** Representative of 3 or more biological replicates. Source data are provided as a Source Data file.

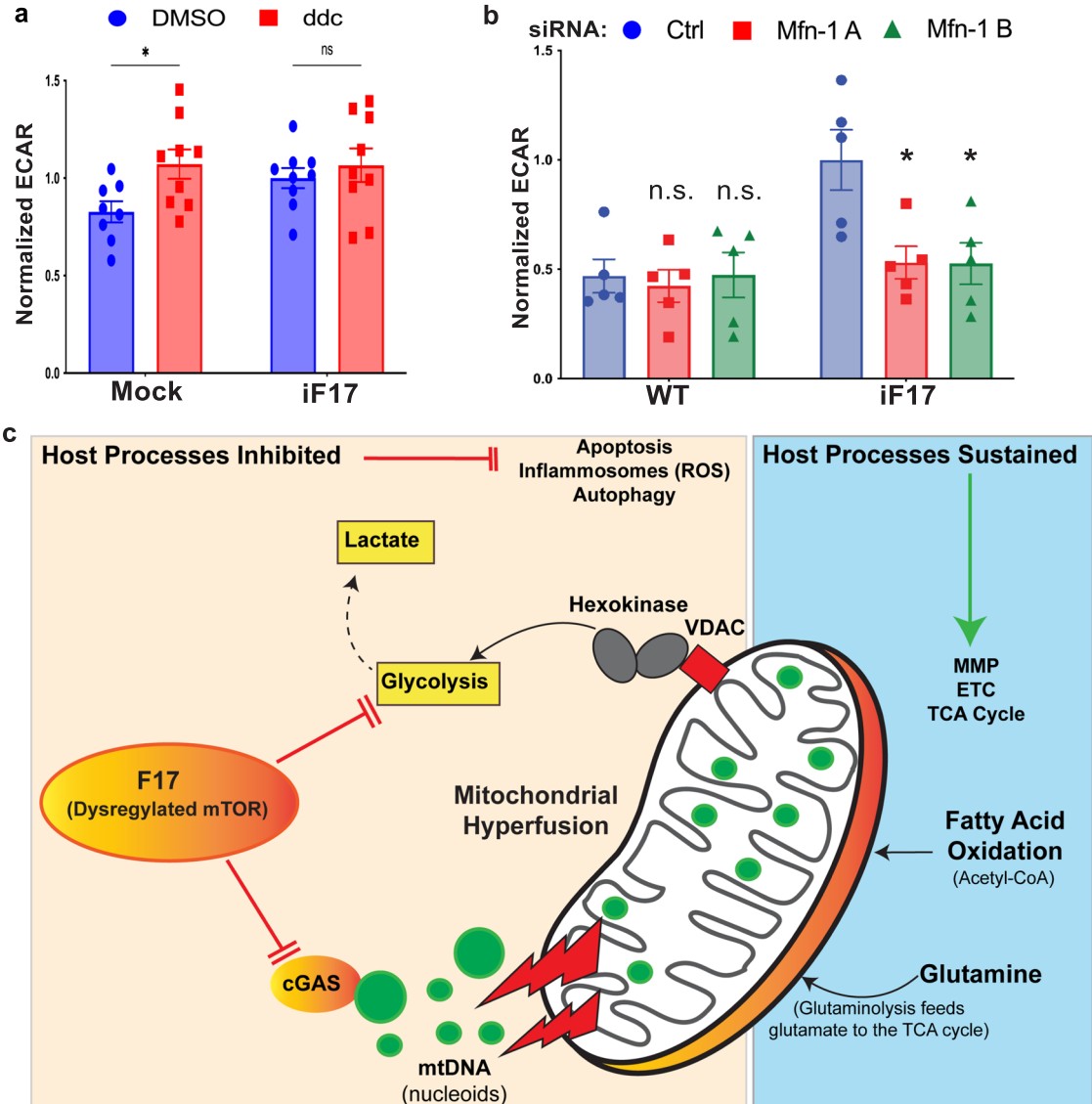

**Fig. 9 | Mitochondrial hyperfusion regulates glycolytic responses to iF17 infection. a** NHDFs were treated with 25 µM ddC prior to infection and measurement of ECAR. n = 8–9 per group, independent two-tailed t-tests. Data are presented as mean values ± SEM. **b** NHDFs were treated with control or MFN1 siRNAs as in Fig. 6a prior to infection and measurement of ECAR. n = 5 per group, multiple independent two-sided *t* tests with the corresponding control siRNA. Data are presented as mean values ± SEM. For (**a, b**); ns = no significance, *p ≤ 0.05. **c** Illustration of host processes that are inhibited or sustained during poxvirus

infection. Poxviruses sustain mitochondrial respiration and fuel the TCA cycle through glutaminolysis and fatty acid oxidation, while blocking glycolysis to inhibit antiviral responses. Mitochondrial fusion plays a central role in orchestrating antiviral responses by leaking mtDNA to activate cGAS and increasing glycolysis to support ISG production. F17 localizes to mitochondria and by targeting mTOR it simultaneously destabilizes cGAS and inhibits glycolysis. Source data are provided as a Source Data file.

under the control of an ISG promoter (ISG54) and different IFN-stimulated response elements. Assays were performed as per the manufacturers instructions. Briefly, the FluostarOMEGA BMG LAB-TECH was primed with Quanti-Luc solution. A 50 µl volume of the

same solution was then injected into each sample well of a white opaque 96-well plate containing 20 µl of cell supernatant. End measurements were made with 4 s start time and 0.1 s reading time.

## Metabolic analyses

Reactive oxygen species levels were measured using the ROS Assay Detection Kit, BioVision Inc/Abcam (Cat #ab287839). Lactate levels were measured using a Lactate Assay Kit, BioVision Inc/Abcam (Cat #ab65330). Protocols were followed as stated by the manufacturer with minor changes. Briefly, for ROS detection cells were seeded at $2.5 \times 10^4$ cells per well onto black opaque 96-well plates with transparent bottoms and infected alongside an uninfected control the following day. Media was then removed and washed with ROS assay buffer. Following this, media was replaced with 100 µl of ROS label in ROS assay buffer and incubated at 37 °C for 45 min in the dark. ROS assay buffer was then used to replace the labeling media and fluorescence was measured immediately. A cellular ROS inducer provided in the kits was used as a positive control and cells devoid of inducer or labeling were used as a negative control. Fluorescence was measured using the FluostarOMEGA BMG LABTECH at Ex/Em = 485/520 nm, and the background measurements were subtracted for final fluorescent figures. Intracellular and extracellular lactate was detected from cells seeded at around 80–90% confluency on a 12-well plate. After infections, media was changed after 22.5 h to 0.5 ml Seahorse XF DMEM medium, pH 7.4, Agilent (Cat# 103575-100) and incubated at 37 °C without $CO_2$ for 1 h. The Agilent medium was then taken off and replaced with 300 µl of the same medium, and after 30 min the sample medium was collected for analysis. Cells were also trypsinized in 100 µl trypsin and collected in 500 µl medium followed by pelleting at 1000 rpm for 4 min. The pellet was lysed in 100 µl ice-cold PBS by 3x flash freeze-thaw cycles and then the cell debris was removed by centrifugation at 1000 rpm for 4 min. The remaining supernatant was taken for analysis. A 20 µl volume of test lysates and supernatants were added to white opaque 96-well plates with 30 µl of lactate assay buffer, followed by 50 µl of assay buffer plus enzyme mix. A negative control devoid of enzyme was also used. To prepare a standard curve and positive control, different volumes of the supplied standard were added to the plate and the volumes were adjusted to 50 µl using assay buffer and then 50 µl of assay buffer plus enzyme mix was added to make 100 µl total. All samples were incubated for 30 min at room temperature in the dark and OD 570 nm or fluorescence at Ex/Em = 544/590 nm was measured using the FluostarOMEGA BMG LABTECH. Calculations for sample concentrations are described in the protocols provided by the manufacturer.

For Oxygen Consumption Rate (OCR) and Extracellular Acidification Rate (ECAR) analyses, sample plates of Seahorse XF96 PDL Cell Culture Microplates (Agilent, Cat# 103730-100) were seeded with approximately $3 \times 10^4$ NHDF cells per well. Approximately one hour before Seahorse analysis, media was removed from the 96-well sample plates and Seahorse XF DMEM medium, pH 7.4, Agilent (Cat# 103575-100) supplemented with 10 mM Seahorse XF 1 M glucose Solution, Agilent (Cat# 103577-100), 2 mM Seahorse XF 200 mM glutamine solution, Agilent (Cat# 103579-100) and 1 mM Seahorse XF 100 mM pyruvate solution, Agilent (Cat# 103578-100) was added. The 96-well sample plates were then incubated at 37 °C without $CO_2$ for 1 h. To prepare for sample analysis, on the prior day Seahorse XFe96 Extracellular Flux Assay Kit Cassettes (Agilent, Cat# 103729-100) were calibrated by adding $R_o$ $H_2O$ to each well and incubating overnight at 37 °C without $CO_2$. During the test sample incubation period, the $H_2O$ was removed from the equilibrated cassette and replaced with Seahorse XF Calibrant Solution, Agilent (Cat# 100840-000), and returned to the 37 °C incubator without $CO_2$ for 1 h. Following the incubation period for the 96-well test plate, media was removed from each well and replaced with supplemented Agilent Seahorse DMEM. Using the XF Cell Energy Phenotype Test Kit, Agilent (Cat# 1033225-100), the cassette and test plates were set up as described in the Agilent protocol for baseline readings of OCR and ECAR using the Seahorse XFe96 Analyzer.

## Densitometry and statistical measurements

Densitometry measurements were performed on autoradiographs using the Fiji distribution of ImageJ[76] (https://fiji.sc/). The background was removed and standardized against a loading control. Where indicated, the band density for uninfected samples was used along with the loading control to generate relative density values. All graphical and statistical analyses were performed using GraphPad Prism from at least 3 independent replicates of each experiment. The statistical tests applied to each experimental setup are detailed in the relevant figure legends.

## Reporting summary

Further information on research design is available in the Nature Portfolio Reporting Summary linked to this article.

## Data availability

The authors declare that the data supporting the findings of this study are available within the article and its Supplementary Information files. Source data are provided with this paper.

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

## Acknowledgements

We thank Bernard Moss, Paula Traktman, Jingxin Cao, Yan Xiang and David Evans for kindly providing reagents. This work was supported by funding from the National Institutes of Health, Grant Numbers P01 HL154998 and P01 AG049665 to N.S.C., and Grant numbers R01 AI127456 and R01 AI179744 to D.W. R.P.C. was supported by a Northwestern University Pulmonary and Critical Care Cugell predoctoral fellowship.

## Author contributions

N.M., N.S.C. and D.W. designed experiments. N.M., H.K.T. R.P.C., C.R.H. and C.P. performed research and analyzed data. N.M. and D.W. wrote the paper. All authors contributed to reading and approving the final version of the manuscript.

## Competing interests

The authors declare no competing interests.
