## [Peer Review File · Nature Communications]

The poxvirus F17 protein counteracts mitochondrially orchestrated antiviral responsesReviewers' Comments:

Reviewer #1:

Remarks to the Author:

The study by Meade et al. proposes that the poxvirus F17 protein localizes to mitochondria to regulate mtDNA release, mitochondrial dynamics, and energy metabolism as a survival strategy to evade the host innate immunity. Although the ideas presented are intriguing, many conclusions are based on limited evidence and several key experiments are lacking proper controls, lessening the overall impact of the study. The authors should address the following points to strengthen the paper.

Major points:

1. One novel aspect proposed in this study is that the viral protein F17 localizes to mitochondria to regulate mtDNA-dependent IFN-I responses. However, this idea is not fully supported by the evidence provided in the manuscript. First, it is unclear whether the phosphorylation of F17 at S53 and S62 are necessary for its mitochondrial localization. The authors should compare Flag-tagged F17 and Flag-tagged F17 (S53,62E) by immunofluorescence staining as they did in Fig. 1C. In addition, Fig. 1D shows the presence of F17 in the Nuc/VF fraction, which potentially contains other organelles. Can the authors probe phospho-F17 in different cell fractions? Second, Fig. 2C-2D show that both wt and iF17 viruses cause mtDNA release, suggesting no role for F17 in regulating mtDNA release. Therefore, the mechanism by which F17 suppresses mtDNA-dependent immune activation is probably by destabilizing cGAS, a finding that has already been reported by this group.
2. Many assays were performed using mock versus iF17 virus infected cells; however, a better comparison would be between wt and iF17 virus infected samples, as other viral factors could significantly alter innate immune outcomes during infection. The authors need to show that WT and iF17 viruses have the same lifecycle, namely that they enter, replicate, and get released in the same way, and then repeat the major findings shown in mock infected cells with WT virus infection.
3. The authors propose that F17 regulates mitochondrial dynamics during infection, presumably by suppressing mitochondrial hyperfusion, to prevent inflammatory glycolysis. However, no evidence in the manuscript convincingly demonstrates that F17 alters mitochondrial morphology. The authors need to provide evidence comparing mitochondrial dynamics during WT vs iF17 infection. They could quantify mitochondrial length by immunofluorescence or probe the activation of proteins that control mitochondrial dynamics (p-Drp1, for example). Moreover, although knockdown of MFN1 reduced heightened glycolysis in iF17 infected cells, the glycolytic status after MFN1 knockdown was not shown in WT virus infected cells. Finally, if F17-regulated mitochondrial dynamics is linked to mtDNA release, how would the authors explain the presence of cytosolic mtDNA in both WT and iF17 infected cells? Does knockdown of MFN1 also decrease cytosolic mtDNA abundance during WT virus infection?
4. The observation that glycolytic inhibition reduces IFN-I responses induced by iF17 infection is intriguing. The authors should validate this conclusion by using genetic approaches to directly target glycolytic enzymes. These approaches may also provide more molecular details on exactly how glycolysis modulates IFN-I responses during poxvirus infection. It has been reported that glycolysis can drive STING activation (<https://www.jci.org/articles/view/166031>). Can the authors probe p-STING with or without glycolysis inhibition during iF17 infection?

Minor points:

1. The authors should add descriptions of the infection time points in each figure, as there is a clear difference between early and late infection shown in Fig. 2B.
2. Figure legend for Fig. 2B should be added.
3. Could the authors clarify their logic for using an anti-ssDNA antibody to stain mtDNA and viral DNA?
4. Antibodies and colors need to be indicated in Extended Data Fig.1A.

Reviewer #2:

Remarks to the Author:

In this work, the authors attempt to demonstrate that disruption of the mitochondrial network will coordinate distinct aspects of the antiviral response following infection with poxviruses. Although poxviruses maintain important mitochondrial functions, such as membrane potential and mitochondrial respiration, reactive oxygen species, a precursor of inflammation, are reduced. Infection and subsequent replication of poxviruses leads to mitochondrial hyperfusion, which induces the release of mitochondrial DNA (mtDNA). This mitochondrial DNA released into the cytosol will lead to an increase in glycolysis, necessary to support the production of interferon-stimulated genes (ISG). On the other hand, the authors highlighted the role of the poxvirus F17 protein, which is localized in the mitochondria and disrupts the action of mTOR, destabilizing cGAS.

This article is difficult to understand and confusing overall. Through the writing, the authors should try to make the article more understandable. Many parts of the "results" section are mixed with discussion, and the reader ends up losing the main thread of the work carried out.

In this work, the authors focus exclusively on the implication of cGAS as an intracytoplasmic sensing molecule for the release of mitochondrial DNA. They did not explore the possible involvement of other cytosolic sensors (AIM2, TLT9, RNA polymerase III). What is the relative ranking of cGAS in relation to the other sensors listed? No experiments using cGAS inhibitors or cGAS siRNA or cGAS KO cells were carried out. The conclusions are too straightforward and there is insufficient experimental evidence to support them.

What is the involvement of poxvirus DNA versus mitochondrial DNA in ISG stimulation? Finally, we do not know (or has not been demonstrated) whether altered poxviruses in the cytosol could release their own DNA that may stimulate cGAS? We also don't know the quantity nor the proportion of poxvirus DNA and mitochondrial DNA. It would be interesting to quantify the proportion of poxvirus DNA and mitochondrial DNA in the purified cytosol.

Line 175 and Figure 2c, the authors measure the quantity of mitochondrial DNA by RT-qPCR, why should mitochondrial RNA production be carried out?

The authors used dideoxycytidine to deplete mitochondrial DNA (Figure 3a). These results should be validated by other approaches such as treating the cells with ethidium bromide, which also leads to mitochondrial DNA depletion.

On the other hand, although mitochondrial DNA is present in the cytosol, we have no evidence of a mechanism leading to the release of mitochondrial DNA. What is the involvement of F17 or another viral protein in this mechanism? Or, is there cooperation between viral and cellular proteins that allows the release of mitochondrial DNA?

Reviewer #3:

Remarks to the Author:

The manuscript from Meade et al describes a well-designed and thorough study dissecting: i) firstly, the specific contribution of mtDNA release to DNA sensing activation and corresponding antiviral host response during poxvirus infections. ii) Secondly, a fine molecular dissection of the viral mechanism, driven by poxvirus F17 protein, to specifically counteract this mtDNA-induced host response. The present work is consequence of two previous studies by the authors related to the role of viral protein F17 on poxvirus evasion of cytosolic sensing through mTOR targeting (Meade et al 2018 Cell; Meade et al 2019 J Virol). In my opinion, the findings from this work sound groundbreaking and provide new insights in a question still hanging in the air in the field of DNA sensing and viral infections: the idea

that mtDNA in the cytosol is a “natural” consequence of infection rather than just a mere consequence of apoptosis at the end of the virus cell-cycle.

However, I have some doubts to be clarified by authors and a few recommendations hoping to improve the MS.

a) The presence of mtDNA in the cytosol is shown at early and very late times in infection (Figure 2). Is 24 hpi when mtDNA is first detected? Have the authors examined this at intermediate times post infection, such as 12 hpi?

I appreciate the use of primary cells (NHDFs) and I wonder whether release of mtDNA during vaccinia infection is cell type dependent or not. Have the authors detected mtDNA in cytosol from infected cells other than NHDFs?

b) According to Figure 2c the levels of mtDNA from cytosolic fractions seem to be much lower in iF17-R compared to iF17 and even to WT. However, in Figure 2d, the qPCR detection shows similar levels (despite SD) for these three viruses. and the authors state in the text (lines 175-6) that cytosolic mtDNA levels increased 4-5-fold in cells infected with either WT, iF17 or iF17R viruses. I can't find an explanation for this discrepancy other than a mistake in Figure 2c. Please clarify.

c) The participation of other poxvirus proteins to prevent the activation of cytosolic detection is not deeply discussed. Proteins such as poxin (degrading cGAMP) or the recently E5 (degrading cGAS) might also prevent the antiviral host response in the absence of F17 at 24 hpi (as occurs at 6hpi), but they do not. These inhibitors prevent cytosolic detection of viral DNA, but they might also contribute to downstream mitigate the mtDNA induced response. In my opinion this topic deserves more discussion.

d) Figure 6.B. I would recommend the authors to remove one of the VBIT-4 concentrations used, since the IFN response seems the same in both cases.

Typos: line 260, “vF1R” refers to vF17R.

We thank the Reviewers for their supportive comments and insightful suggestions. We have addressed all comments directly, added extensive new data and carefully edited the manuscript in light of the comments from all three Reviewers. Notably, this involved moving several text sections from our results to more appropriate places in either the introduction or discussion, which removed repetition and greatly improved the flow and focus of the revised manuscript. Combined with the new data added, we feel that this has greatly improved the quality of this manuscript and further strengthened support for our original findings and model. We hope that our revised manuscript and responses below address the Reviewer concerns satisfactorily.

Reviewer #1 (Remarks to the Author):

The study by Meade et al. proposes that the poxvirus F17 protein localizes to mitochondria to regulate mtDNA release, mitochondrial dynamics, and energy metabolism as a survival strategy to evade the host innate immunity. Although the ideas presented are intriguing, many conclusions are based on limited evidence and several key experiments are lacking proper controls, lessening the overall impact of the study. The authors should address the following points to strengthen the paper.

Major points:

1. One novel aspect proposed in this study is that the viral protein F17 localizes to mitochondria to regulate mtDNA-dependent IFN-I responses. However, this idea is not fully supported by the evidence provided in the manuscript. First, it is unclear whether the phosphorylation of F17 at S53 and S62 are necessary for its mitochondrial localization. The authors should compare Flag-tagged F17 and Flag-tagged F17 (S53,62E) by immunofluorescence staining as they did in Fig. 1C. In addition, Fig. 1D shows the presence of F17 in the Nuc/VF fraction, which potentially contains other organelles. Can the authors probe phospho-F17 in different cell fractions?

We appreciate the Reviewers point and thank them for this insightful suggestion. Because WT F17 exists in both phosphorylated and non-phosphorylated forms that we can neither control nor distinguish, we performed a variation on the suggested experiment by expressing a Flag-tagged non-phosphorylatable Serine-to-Alanine (S53/62A) form of F17. As we now show in Supplementary Fig. 3, non-phosphorylated F17 localizes to viral factories and forms speckled structures commonly reported for viral structural proteins. This also aligns with studies from other labs cited in this manuscript which report that phosphorylation of F17 is not involved in its virion morphogenesis functions. Combined with our results using the phosphomimetic form of F17, this new data further supports prior suggestions that phosphorylation partitions F17's dual functions in morphogenesis and immune evasion.

With regard to the Reviewer's second query of whether we can also probe fractions for phospho-F17, unfortunately we have tried several times to make phospho-specific antibodies but with no success. Furthermore, we agree with the Reviewer's related comment that these fractions can still contain other organelles, and therefore has its limitations. Indeed, on hindsight, we clearly cannot separate a large proportion of mitochondria that tether to other organelles from co-sedimenting with nuclei and viral factories (VFs). As such, we have moved the fractionation data to a supplemental figure (Supplementary Fig. 2) and simplified our

conclusions from this particular approach, which we now use to simply provide additional evidence of F17's mitochondrial localization. We feel that the Reviewer's primary suggestion of imaging F17 localization addressed this question more directly.

We have modified the text accordingly and briefly mention that while we hope to further explore the mechanisms that regulate F17's mitochondrial localization in the future, the basic discovery that phosphorylated F17 localizes to mitochondria provided the first clues for us to subsequently uncover F17's functional role in counteracting mitochondrially orchestrated antiviral responses. We hope that this new data and modifications to the revised manuscript address the Reviewer's concern satisfactorily.

Second, Fig. 2C-2D show that both wt and iF17 viruses cause mtDNA release, suggesting no role for F17 in regulating mtDNA release. Therefore, the mechanism by which F17 suppresses mtDNA-dependent immune activation is probably by destabilizing cGAS, a finding that has already been reported by this group.

We respectfully argue that our prior discovery that F17 destabilizes cGAS was just the tip of the iceberg in terms of understanding how F17 counteracts innate responses, while the precise nature of these responses was entirely unknown. This report therefore contains several novel findings and advances: first, we agree, F17 does not regulate mtDNA release but that in-itself is novel, as targeting the downstream events is distinct from viruses like HSV-1 that directly degrade mtDNA. We elaborate on this, along with the fact that there is growing evidence that mtDNA release can be a specific cellular response to several viruses in the revised manuscript. Second, our prior report, like many others, assumed viral DNA was the cGAS agonist. We now show that mtDNA plays an unexpectedly important role in this process. Third, we further reveal a novel role for mitochondrial hyperfusion in enabling an increase glycolysis, itself a distinct but important driver of antiviral responses. Fourth, we show that destabilizing cGAS is just part of the mechanism by which F17 counteracts host responses and that it also prevents host increases in glycolysis. As we discuss in more detail in the revised manuscript, this would allow F17 to broadly impair host responses to a wide range of stimuli, notably including but not limited to mtDNA. Moreover, our findings also explain the reason why, as prior reports have shown, VacV does not activate glycolysis but instead uses glutaminolysis and fatty acid oxidation to support the TCA cycle. We respectfully feel that these findings represent significant and novel advances on multiple levels.

2. Many assays were performed using mock versus iF17 virus infected cells; however, a better comparison would be between wt and iF17 virus infected samples, as other viral factors could significantly alter innate immune outcomes during infection. The authors need to show that WT and iF17 viruses have the same lifecycle, namely that they enter, replicate, and get released in the same way, and then repeat the major findings shown in mock infected cells with WT virus infection.

We appreciate the Reviewers point and we agree that this is particularly important in the context of mitochondrial hyperfusion and its role in increasing glycolysis in response to infection. We acknowledge that there is limited hyperfusion in uninfected cells and as such, MFN1 depletion may have a limited impact in this context, while it may have effects in WT-

infected cells wherein hyperfusion is occurring. As such, we have repeated MFN1 knockdowns in WT infected cells. New data in Figure 9b and Supplementary Fig. 8 show that, similar to mock-infected cells, MFN1 depletion has no significant effect on glycolysis despite reducing mtDNA release. This further strengthens our conclusion that mitochondrial hyperfusion allows the host to increase glycolysis when the viral F17 antagonist is absent.

Beyond MFN1 depletion, we had in fact considered the Reviewer's broader point from the outset of this project. We respectfully first point out that for all key processes, inhibitors or metabolic adaptation conditions, we do in fact compare all four conditions of mock, WT, iF17 and iF17R, and no effects are observed for WT or iF17R. We only focus in on mock versus iF17 infection to explore the mechanisms behind these responses in more detail. In some cases, the experiments simply demonstrate that there is no role for a process in the response to iF17, so a WT comparison is uninformative and respectfully, not necessary. In others, we chose a mock control rather than a WT comparison for our follow-up functional testing for a number of reasons. First, while we agree that other viral proteins affect immune responses, they are also expressed by iF17 and the question then becomes, what exactly are we trying to measure or compare here? In initial experiments we did include WT and iF17R infections but because they inhibit the process we are measuring, unsurprisingly we saw no effects. In the interests of time, money and figure space, we omitted them from replicate experiments as they were not informative. This choice was also driven by the fact that WT/iF17R infection actually downregulates ISG expression below basal levels in uninfected cells. This is clearly highlighted in the quantification of ISG responses presented in Figure 2d as well as other data throughout the revised manuscript. As a result, using WT infections as our comparative baseline to iF17 would artificially inflate the magnitude of apparent responses because, at least in our mind, we would not be measuring host responses (changes from the uninfected state) but instead, we would be comparing activation of host responses against a repressed state. Ultimately, the choice of mock versus WT comparison does not affect the results themselves, beyond amplifying magnitudes. While this is somewhat of an intellectual argument in which both views have validity, we respectfully argue that in our view the most informative comparison is actually how the uninfected cell changes when it responds, namely mock versus iF17. We hope this this new data and our responses address the Reviewer's concerns satisfactorily.

With regards to the Reviewer's comment on the need to compare the viral lifecycles and show they are the same, this is not necessarily going to be true given that the iF17 mutant can't control mTOR and elicits an antiviral response. However, several studies cited in this manuscript have shown that WT and iF17 replication is the same in various cell lines, with the notable exception of virion morphogenesis. In line with this, our previously published data along with new data throughout this manuscript show that the expression of early and late genes is unaffected (e.g. Supplementary Fig. 6), both in terms of levels and kinetics, ruling out entry or gene expression defects. Notably, we do detect small reductions in viral DNA replication (Supplementary Fig. 5) which, as we note in the revised manuscript, may reflect either requirements for mTOR dysregulation in supplying nucleotide pools or perhaps even an effect of the host antiviral response that is mounted against iF17. However, while future studies will explore this in more detail, the effects are small and an important point for this report is

that there is no increase in viral DNA that might explain the ISG responses observed during iF17 infection. In terms of virus release, the bulk of VacV particles are intracellular and as reported by others, and noted in the manuscript, viral particles form but do not mature to become infectious in the absence of F17. As such, we cannot measure production of virus particles by traditional plaque assays. However, our data clearly shows that none of these differences contribute to host responses to iF17 infection because the core processes of mitochondrial hyperfusion and mtDNA release, which initiate these responses, occur in WT, iF17 or iF17R infected cells, and blocking mtDNA release, hyperfusion or glycolysis block responses in the absence of the viral inhibitor, F17. We have modified the text to discuss these points in the revised manuscript, and we hope that our modifications and responses have addressed the Reviewer's concern satisfactorily.

3. The authors propose that F17 regulates mitochondrial dynamics during infection, presumably by suppressing mitochondrial hyperfusion, to prevent inflammatory glycolysis. However, no evidence in the manuscript convincingly demonstrates that F17 alters mitochondrial morphology. The authors need to provide evidence comparing mitochondrial dynamics during WT vs iF17 infection. They could quantify mitochondrial length by immunofluorescence or probe the activation of proteins that control mitochondrial dynamics (p-Drp1, for example). Moreover, although knockdown of MFN1 reduced heightened glycolysis in iF17 infected cells, the glycolytic status after MFN1 knockdown was not shown in WT virus infected cells. Finally, if F17-regulated mitochondrial dynamics is linked to mtDNA release, how would the authors explain the presence of cytosolic mtDNA in both WT and iF17 infected cells? Does knockdown of MFN1 also decrease cytosolic mtDNA abundance during WT virus infection?

With regard to the Reviewer's general comments about F17's role in regulating mitochondrial morphology, we respectfully point out that we do not claim that F17 plays any role in these specific events. Our data shows that F17 blocks the downstream consequences of changes in mitochondrial hyperfusion and mtDNA release, which occur in response to infection with either WT or iF17 viruses. Moreover, similar changes have been reported for HSV-1 and Measles Virus (MeV). As such, the changes in mitochondria that the Reviewer refers to are driven not by F17 but rather as a cellular response to infection itself. We hope that we have clarified this point in the revised manuscript.

With regard to the Reviewer's suggestion to examine effects on mitochondrial dynamics in more detail, we thank them for suggesting these two potential approaches. The extent of mitochondrial hyperfusion makes it extremely difficult to reliably measure mitochondrial length by imaging. As such, we focused on the alternative approach suggested by the Reviewer and we now show that phosphorylation of DRP1 at Serine 616 is reduced upon infection, which would impair fission activity. In addition, we find that phosphorylation of MFN2 at Serine 422 is also reduced, which is an inhibitory modification and would increase fusion activity. As such, our new data shows that infection skews the fission-fusion balance towards fusion and combined with reductions in PINK1, these findings would explain why we see extensive mitochondrial hyperfusion in response to infection, irrespective of the presence or absence of F17. We thank the Reviewer for suggesting these experiments and we hope that our responses and revisions address this concern satisfactorily.

4. The observation that glycolytic inhibition reduces IFN-I responses induced by iF17 infection is intriguing. The authors should validate this conclusion by using genetic approaches to directly target glycolytic enzymes. These approaches may also provide more molecular details on exactly how glycolysis modulates IFN-I responses during poxvirus infection. It has been reported that glycolysis can drive STING activation (<https://www.jci.org/articles/view/166031>). Can the authors probe p-STING with or without glycolysis inhibition during iF17 infection?

We appreciate the Reviewer's point and we have now independently validated our original findings that were made using inhibitors of glycolysis. As highlighted by their prevalence in Depmap (<https://depmap.org/>), these are essential genes that are difficult to target genetically. As such, we instead used a well-established alternative approach to addressing the Reviewer's comment, wherein we adapted cells from Glucose-containing to Galactose-containing medium. This greatly reduces flux through glycolysis while continuing to support energy needs through mitochondrial oxidative phosphorylation. We demonstrate the efficacy of this switch using Piericidin controls (Fig. 8c of the revised manuscript). Moreover, adapting cells to Galactose significantly reduces ISG responses to iF17 infection (Fig. 8d, e of the revised manuscript), independently confirming the importance of glycolysis. We hope that this alternative approach satisfies the Reviewer's first point.

With regards to roles for glycolysis in STING activation, we probed our inhibitor-treated samples as suggested but found that there was no suppression of phosphorylated STING levels (Fig. 8a of the revised manuscript). Most likely this just reflects broader differences in how STING and glycolysis crosstalk or not during different host responses. Our data suggests that glycolysis and cGAS-STING signaling contribute independently to host responses to VacV infection, which aligns more broadly with our model as to why F17 would target both processes rather than just cGAS-STING signaling alone. We have modified the revised manuscript accordingly and hope that these findings and responses address the Reviewer's concern.

Minor points:

1. The authors should add descriptions of the infection time points in each figure, as there is a clear difference between early and late infection shown in Fig. 2B.

We have added more detail as requested, along with the legend accidentally omitted (comment below). This is now part of a more expanded dataset shown in Supplementary Fig. 6 along with Fig. 2d of the revised manuscript.

2. Figure legend for Fig. 2B should be added.

We apologize for this oversight. This has been corrected (also see point 1 above).

3. Could the authors clarify their logic for using an anti-ssDNA antibody to stain mtDNA and viral DNA?

We apologize for any confusion caused by this labeling. The antibody in question is sometimes referred to as being anti-ssDNA as it seems to have a preference, but in fact, it is raised against DNA. However, it predominantly detects mtDNA nucleoids, as is evident in ours and others imaging approaches, likely due to high levels or concentrations of ssDNA. How the antibody is

referred to in the literature varies and can cause confusion. We thank the Reviewer for raising this point, which we have clarified in the revised manuscript.

4. Antibodies and colors need to be indicated in Supplementary Fig.1A. We apologize for this oversight. This has been corrected.

Reviewer #2 (Remarks to the Author):

In this work, the authors attempt to demonstrate that disruption of the mitochondrial network will coordinate distinct aspects of the antiviral response following infection with poxviruses. Although poxviruses maintain important mitochondrial functions, such as membrane potential and mitochondrial respiration, reactive oxygen species, a precursor of inflammation, are reduced. Infection and subsequent replication of poxviruses leads to mitochondrial hyperfusion, which induces the release of mitochondrial DNA (mtDNA). This mitochondrial DNA released into the cytosol will lead to an increase in glycolysis, necessary to support the production of interferon-stimulated genes (ISG). On the other hand, the authors highlighted the role of the poxvirus F17 protein, which is localized in the mitochondria and disrupts the action of mTOR, destabilizing cGAS.

This article is difficult to understand and confusing overall. Through the writing, the authors should try to make the article more understandable. Many parts of the "results" section are mixed with discussion, and the reader ends up losing the main thread of the work carried out. In this work, the authors focus exclusively on the implication of cGAS as an intracytoplasmic sensing molecule for the release of mitochondrial DNA. They did not explore the possible involvement of other cytosolic sensors (AIM2, TLT9, RNA polymerase III). What is the relative ranking of cGAS in relation to the other sensors listed? No experiments using cGAS inhibitors or cGAS siRNA or cGAS KO cells were carried out. The conclusions are too straightforward and there is insufficient experimental evidence to support them.

We appreciate the Reviewer's viewpoint. In retrospect, having taken a step back from the manuscript writing process, we realize that we did indeed mix too much of what should be in our introduction or discussion into the results section, in several places we got into excessive details, while we also repeated points in the results and discussion section. We have extensively edited the revised manuscript to address these issues and we hope that our changes address the Reviewer's concern.

With regards to comments relating to cGAS, we previously tested the role of other sensors and established the central importance of cGAS. We appreciate, however, that this involved other cell types beyond those used in this study. As such, as suggested by the Reviewer, we have now used cGAS inhibitors and cGAS knockouts to demonstrate the essential role of cGAS in both primary NHDFs and THP1 monocytes that are used here. While this of course does not rule out roles for other sensors, particularly those that may cooperate with or require cGAS, we hope that our new data now clearly supports the central importance of cGAS and the basis of our overall conclusions. We have modified the revised manuscript to reflect these points.

What is the involvement of poxvirus DNA versus mitochondrial DNA in ISG stimulation? Finally, we do not know (or has not been demonstrated) whether altered poxviruses in the cytosol could release their own DNA that may stimulate cGAS? We also don't know the quantity nor the proportion of poxvirus DNA and mitochondrial DNA. It would be interesting to quantify the proportion of poxvirus DNA and mitochondrial DNA in the purified cytosol.

The Reviewer raises some interesting points and ideas. To address these, we now include new data measuring viral DNA replication and viral DNA levels in the cytosol (Supplementary Fig. 5). Notably, prior studies have shown that the absence of F17 results in defects in poxvirus morphogenesis, but no evidence of inadvertent release of viral DNA. In line with this, we do not find any increase in viral DNA in the cytosol. In fact, despite no effects on viral gene expression (also reported by others), we do observe a small decrease in viral DNA replication and total viral DNA in iF17-infected cells that may reflect either the need to dysregulate mTOR to supply nucleotide pools or even potential effects of the antiviral responses that this mutant fails to block. We now briefly discuss this point along with the fact that regardless of the cause, this finding also rules out the possibility that increased levels of cytosolic viral DNA might explain the ISG responses that we observe during iF17 infection. Of course, we clearly demonstrate that mtDNA contributes to these responses regardless, but we feel that this is an important addition, and we thank the Reviewer for suggesting this experiment.

Related to this and the Reviewer's other question regarding the relative contribution of viral DNA versus mtDNA, it is challenging, if not impossible, to quantitatively compare these different DNA species or interpret the meaning of any result. Even if we could accurately compare relative levels, we do not know if mtDNA or vDNA are localized more optimally or are qualitatively better agonists for sensors such as cGAS, so direct comparisons are hard to make. However, we do recognize that our approaches that suppress mtDNA levels or release do not absolutely inhibit host responses, suggesting that other DNA species such as viral DNA do indeed contribute to the overall activation of cGAS. Furthermore, based on the Reviewer's comment regarding EtBr (see response to comment two points below), we have modified our discussion to acknowledge the likely contributions from viral DNA, while highlighting the novel roles played by mtDNA and mitochondrial hyperfusion in driving cellular responses that form the focus of this report. Moreover, the multifunctionality of F17 would explain why it is so important in blocking responses to multiple forms of DNA, including the unexpectedly important role of mtDNA in eliciting responses to a cytoplasmic DNA virus. We have modified our results and discussion to reflect these points.

Line 175 and Figure 2c, the authors measure the quantity of mitochondrial DNA by RT-qPCR, why should mitochondrial RNA production be carried out?

We apologize for any confusion here. The "RT" does not refer to Reverse Transcription but rather, Real Time quantitative PCR. We have clarified this in the revised manuscript.

The authors used dideoxycytidine to deplete mitochondrial DNA (Figure 3a). These results should be validated by other approaches such as treating the cells with ethidium bromide, which also leads to mitochondrial DNA depletion.

We thank the Reviewer for suggesting this experiment as the results were quite interesting and relate back to the Reviewer's prior comment (two above). We performed the suggested experiment and added new data showing that EtBr treatment results in a very robust suppression of ISG responses to infection (Supplementary Fig. 7). However, because EtBr not only depletes mtDNA but also intercalates into DNA indiscriminately, this may affect the quantity and/or quality of other forms of DNA, such as viral DNA or even host genomic DNA, as agonists for immune responses. As such, we show this data but interpret it with caution, highlighting that our use of the more specific inhibitor, ddC as well as MFN1 depletion to directly control mtDNA levels or release, does not cause the same magnitude of effects. This takes us back to the Reviewer's earlier comment, and we have modified our text and conclusions to acknowledge likely roles for other forms of DNA in driving the overall response, while highlighting the specific contribution from mtDNA through our more targeted approaches.

On the other hand, although mitochondrial DNA is present in the cytosol, we have no evidence of a mechanism leading to the release of mitochondrial DNA. What is the involvement of F17 or another viral protein in this mechanism? Or, is there cooperation between viral and cellular proteins that allows the release of mitochondrial DNA?

We now show that infection reduces phosphorylation of the fission factor, DRP1 and reduces PINK1 levels, while also reducing inactivating phosphorylation of the fusion factor MFN2 (Fig. 2c). Combined, this would impair fission and clearance of mitochondria, while increasing fusion, explaining why infection results in mitochondrial hyperfusion. Mitochondrial hyperfusion is a controlled process that leads to mtDNA release that is also observed during infection with HSV-1 or Measles Virus (MeV). Moreover, we also observe hyperfusion and mtDNA release in WT or iF17 infections. Combined, this strongly suggests that this is a host response to infection that is not directly controlled by the virus. This also addresses the Reviewer's second question regarding the role of F17 or other viral proteins, the answer being that viral proteins may only play an indirect role, if any in triggering the cell to mount responses to virus infection. As our data shows, F17 does not cause or control mitochondrial hyperfusion or mtDNA release, but instead it counteracts the downstream consequences that otherwise lead to innate responses to infection. We hope that our revisions and responses address the Reviewer's comments satisfactorily.

Reviewer #3 (Remarks to the Author):

The manuscript from Meade et al describes a well-designed and thorough study dissecting: i) firstly, the specific contribution of mtDNA release to DNA sensing activation and corresponding antiviral host response during poxvirus infections. ii) Secondly, a fine molecular dissection of the viral mechanism, driven by poxvirus F17 protein, to specifically counteract this mtDNA-induced host response. The present work is consequence of two previous studies by the authors related to the role of viral protein F17 on poxvirus evasion of cytosolic sensing through mTOR targeting (Meade et al 2018 Cell; Meade et al 2019 J Virol). In my opinion, the findings from this work sound groundbreaking and provide new insights in a question still hanging in the air in the field of DNA sensing and viral infections: the idea that mtDNA in the cytosol is a

“natural” consequence of infection rather than just a mere consequence of apoptosis at the end of the virus cell-cycle.

However, I have some doubts to be clarified by authors and a few recommendations hoping to improve the MS.

a) The presence of mtDNA in the cytosol is shown at early and very late times in infection (Figure 2). Is 24 hpi when mtDNA is first detected? Have the authors examined this at intermediate times post infection, such as 12 hpi?

We appreciate the Reviewers point and thank them for this suggestion. Although, we would point out that 24h is not actually very late and that further increases in both viral protein production and replication continue through 48h in primary cells (e.g. Hesser et al, J Virol, 2023). Regardless, the suggested experiment was very informative. We analyzed mtDNA levels along with corresponding responses to infection at 12hpi and 24h.p.i. These new data, shown in Figures 2b, 2d and Supplementary Fig. 6 of the revised manuscript, demonstrate that mtDNA release becomes detectable at this intermediate timepoint and is further increased at 24h.p.i. Moreover, the kinetics of mtDNA release correlate with the increases in ISG responses over time, clearly demonstrating that both processes correlate with one another and with the progression of the virus replication cycle after it is established.

I appreciate the use of primary cells (NHDFs) and I wonder whether release of mtDNA during vaccinia infection is cell type dependent or not. Have the authors detected mtDNA in cytosol from infected cells other than NHDFs?

We now show that cytosolic mtDNA levels increase during infection and that ddC treatment reduces ISG responses in THP1 monocytes (Supplementary Fig. 4b and Fig. 4e-f of the revised manuscript). In unpublished experiments we have also detected increases in cytosolic mtDNA in lung fibroblasts, further suggesting that is not a cell type specific event.

b) According to Figure 2c the levels of mtDNA from cytosolic fractions seem to be much lower in iF17-R compared to iF17 and even to WT. However, in Figure 2d, the qPCR detection shows similar levels (despite SD) for these three viruses. and the authors state in the text (lines 175-6) that cytosolic mtDNA levels increased 4-5-fold in cells infected with either WT, iF17 or iF17R viruses. I can't find an explanation for this discrepancy other than a mistake in Figure 2c. Please clarify.

We apologize for any confusion caused by this panel. While our RT-qPCR is quantitative and carefully normalized we clearly picked a poor example of conventional PCR runs, largely meant to provide a visual of the mtDNA release and show primer specificity, as the cytosolic fraction of the iF17R sample was underloaded in this particular case. We have rerun these samples and replaced the affected panels to avoid any confusion for readers.

c) The participation of other poxvirus proteins to prevent the activation of cytosolic detection is not deeply discussed. Proteins such as poxin (degrading cGAMP) or the recently E5 (degrading cGAS) might also prevent the antiviral host response in the absence of F17 at 24 hpi (as occurs at 6hpi), but they do not. These inhibitors prevent cytosolic detection of viral DNA, but they might also contribute to downstream mitigate the mtDNA induced response. In my opinion this

topic deserves more discussion.

We appreciate the Reviewers point and we have modified our discussion accordingly. In light of this and comments from Reviewer 2, we moved our initial brief discussion of poxin (B2) from the introduction and elaborated on this point in the discussion section. We now discuss how B2 and F17 clearly affect distinct aspects of cGAS activity at distinct phases of infection, how they likely function cooperatively, and how their combined absence makes sense in terms of broader studies of MVA's inability to efficiently block cGAS responses. We also discuss the recent report on E5, which was published during the review of our manuscript. This report was initially quite perplexing to us as not only do we not detect such rapid and extensive degradation of cGAS, but it also seems counterintuitive that MVA encodes such a potent cGAS antagonist based on several prior studies from multiple groups. However, E5 was identified using complicated HEK293-based screens that identified many candidates, while it appears that the effects of deleting the E5-encoding gene (E5R) in the context of infection were only examined in contexts where infection is abortive. As such, while deletion of E5R may provide a valuable means to improve immune responses and vaccine efficacy, it remains to be determined whether it modulates cGAS degradation directly or indirectly, and whether it functions in other cell types and/or during productive infection. We thank the Reviewer for this suggestion. We hope that our revised discussion sets better context for each of the currently reported cGAS inhibitors, which will also help to avoid any confusion over seemingly contradictory results reported in different studies.

d) Figure 6.B. I would recommend the authors to remove one of the VBIT-4 concentrations used, since the IFN response seems the same in both cases.

We have removed the second VBIT-4 concentration as suggested.

Typos: line 260, "vF1R" refers to vF17R.

We thank the Reviewer for spotting this typo. This has been corrected.

Reviewers' Comments:

Reviewer #1:

Remarks to the Author:

The authors have revised the manuscript text significantly and provided clarification for many of the points I raised. New data are provided that strengthen the conclusions, such as the localization of phosphorylated viral protein F17, questions on mitochondrial dynamics, and analysis of the interplay between cGAS-dependent ISG expression and glycolysis-mediated IFN-I responses. The authors have also toned down some of their conclusions. I support publication of the revised paper.

Reviewer #2:

Remarks to the Author:

Although the manuscript has been improved with the addition of new experiments, some of the points previously raised have still not been addressed. Many of these points are still quite confusing.

The authors often performed experiments that had already been carried out or are redundant. Although cGAS has been extensively described and implicated in the interferon pathway, we have no experimental data showing the effect of other sensors such as AIM2, TLT9, RIG-I/RNA polymerase III compared to c-GAS. This question will serve to rank the involvement of the various sensors involved.

The authors compared vDNA by RT-qPCR following infection with the WT, iF17 and iF17-R viruses, concluding that there was no evidence of the presence of vDNA.

The authors regularly respond with this type of statement: "It is challenging, if not impossible, to quantitatively compare these different DNA species or interpret the meaning of any result". To answer this question, it would be worthwhile to be more descriptive and precise. This is possible, for example, by quantifying the proportion of vDNA versus mtDNA by NGS in the purified cytosolic compartment. The authors do not present the controls of cytosol purification.

The experiment demonstrating the depletion of mtDNA following EtBr treatment is not convincing. A 48-hour incubation is not sufficient to observe the depletion of mtDNA. Furthermore, no experiment is proposed to show the amount of mtDNA in the cytosol following EtBr treatment.

Figure 2C and 3C: It is difficult for the reviewer to understand how the authors could leave a figure unfinalized.

Reviewer #3:

Remarks to the Author:

I would like to thank the authors for their efforts to address my previous concerns and suggestions. In my opinion this revision version of the manuscript is easier to read and understand. In particular, I appreciate the discussion around the recently reported degradation of cGAS by the E5 protein. It was also quite shocking for me and I thought it was worth highlighting in the discussion. Also, the authors have extended their findings to other cell type (Thp1 monocytes) and additional times after infection.

Reviewer #1 (Remarks to the Author):

The authors have revised the manuscript text significantly and provided clarification for many of the points I raised. New data are provided that strengthen the conclusions, such as the localization of phosphorylated viral protein F17, questions on mitochondrial dynamics, and analysis of the interplay between cGAS-dependent ISG expression and glycolysis-mediated IFN-I responses. The authors have also toned down some of their conclusions. I support publication of the revised paper.

We thank the Reviewer for their kind and supportive comments.

Reviewer #2 (Remarks to the Author):

Although the manuscript has been improved with the addition of new experiments, some of the points previously raised have still not been addressed. Many of these points are still quite confusing.

The authors often performed experiments that had already been carried out or are redundant. Although cGAS has been extensively described and implicated in the interferon pathway, we have no experimental data showing the effect of other sensors such as AIM2, TLT9, RIG-I/RNA polymerase III compared to c-GAS. This question will serve to rank the involvement of the various sensors involved.

We feel that the Reviewers original suggestion to test the importance of cGAS was valuable as we had not tested its importance in NHDFs or THP1 monocytes previously. Beyond cGAS, as per our prior responses, testing the role of other sensors and somehow ranking them is technically challenging, difficult to interpret and beyond the scope of this report.

The authors compared vDNA by RT-qPCR following infection with the WT, iF17 and iF17-R viruses, concluding that there was no evidence of the presence of vDNA.

The authors regularly respond with this type of statement: "It is challenging, if not impossible, to quantitatively compare these different DNA species or interpret the meaning of any result".

To answer this question, it would be worthwhile to be more descriptive and precise. This is possible, for example, by quantifying the proportion of vDNA versus mtDNA by NGS in the purified cytosolic compartment. The authors do not present the controls of cytosol purification.

We respectfully point out that we do not claim that there is no evidence of the presence of vDNA. Quite the opposite, as we show that there is vDNA present in the cytosol and discuss how it likely also contributes to host sensing. However, this report focuses on the discovery that mtDNA acts as a surprisingly important trigger for host responses to poxvirus infection.

As per our prior responses, NGS analysis would be technically challenging to perform yet would add little or nothing to our understanding given the inability to address qualitative differences in how different DNA species may serve as agonists for sensors. Moreover, our focus is not on comparing DNA species but on revealing the unexpectedly important role of mtDNA in activating host responses to poxvirus infection.

With regards to controls for cytosol purification, we have modified the text in the figure legend for Supplemental Fig. 5c to make it clearer to readers that samples used for vDNA analysis are

the same as those used for mtDNA analysis in Figure 2a,b, wherein cytosolic controls are presented (namely lack of GAPDH in these fractions). Moreover, for imaging analysis of vDNA in viral factories, these are clearly cytosolic structures.

The experiment demonstrating the depletion of mtDNA following EtBr treatment is not convincing. A 48-hour incubation is not sufficient to observe the depletion of mtDNA. Furthermore, no experiment is proposed to show the amount of mtDNA in the cytosol following EtBr treatment.

While longer treatment periods using lower concentrations of EtBr are often used to eliminate mtDNA, many studies use 48-hour incubations with higher concentrations such as those used here in order to deplete mtDNA from cells more transiently. This is preferable in many cases due to the broader effects of EtBr on genomic DNA. While its efficacy was abundantly clear in the magnitude of the effects of EtBr treatment on host responses to infection in the original submission, we have added new data (Supplementary Fig. 7A of the revised manuscript) further showing that 48-hour treatment with EtBr causes a large reduction in the level of mtDNA nucleoids in cells prior to infection. However, while efficacious and conducted at the Reviewer's request, as we stated in our prior responses, EtBr indiscriminately intercalates into diverse DNA species and has broad effects. As such, we cannot make firm conclusions about the role of mtDNA from EtBr-based approaches, which is why we focus on more specific strategies to address this question.

Figure 2C and 3C: It is difficult for the reviewer to understand how the authors could leave a figure unfinalized.

We respectfully point out that neither of these figures are unfinalized. This is common practice for specific controls, and the panels that the Reviewer refers to simply include controls that serve to show that we can detect apoptosis as we are showing a null effect during infection (Fig. 2C) and that the lack of detectable cGAS in cGAS KO cells is true, and not simply due to failure of the Western blot itself (Fig. 3C).

Reviewer #3 (Remarks to the Author):

I would like to thank the authors for their efforts to address my previous concerns and suggestions.

In my opinion this revision version of the manuscript is easier to read and understand. In particular, I appreciate the discussion around the recently reported degradation of cGAS by the E5 protein. It was also quite shocking for me and I thought it was worth highlighting in the discussion.

Also, the authors have extended their findings to other cell type (Thp1 monocytes) and additional times after infection.

We thank the Reviewer for their kind and supportive comments.

Reviewer #1 (Remarks to the Author):

The authors have revised the manuscript text significantly and provided clarification for many of the points I raised. New data are provided that strengthen the conclusions, such as the localization of phosphorylated viral protein F17, questions on mitochondrial dynamics, and analysis of the interplay between cGAS-dependent ISG expression and glycolysis-mediated IFN- α responses. The authors have also toned down some of their conclusions. I support publication of the revised paper.

We thank the Reviewer for their kind and supportive comments.

Reviewer #2 (Remarks to the Author):

Although the manuscript has been improved with the addition of new experiments, some of the points previously raised have still not been addressed. Many of these points are still quite confusing.

The authors often performed experiments that had already been carried out or are redundant. Although cGAS has been extensively described and implicated in the interferon pathway, we have no experimental data showing the effect of other sensors such as AIM2, TLT9, RIG-I/RNA polymerase III compared to c-GAS. This question will serve to rank the involvement of the various sensors involved.

We feel that the Reviewers original suggestion to test the importance of cGAS was valuable as we had not tested its importance in NHDFs or THP1 monocytes previously. Beyond cGAS, as per our prior responses, testing the role of other sensors and somehow ranking them is technically challenging, difficult to interpret and beyond the scope of this report.

The authors compared vDNA by RT-qPCR following infection with the WT, iF17 and iF17-R viruses, concluding that there was no evidence of the presence of vDNA.

The authors regularly respond with this type of statement: "It is challenging, if not impossible, to quantitatively compare these different DNA species or interpret the meaning of any result".

To answer this question, it would be worthwhile to be more descriptive and precise. This is possible, for example, by quantifying the proportion of vDNA versus mtDNA by NGS in the purified cytosolic compartment. The authors do not present the controls of cytosol purification.

We respectfully point out that we do not claim that there is no evidence of the presence of vDNA. Quite the opposite, as we show that there is vDNA present in the cytosol and discuss how it likely also contributes to host sensing. However, this report focuses on the discovery that mtDNA acts as a surprisingly important trigger for host responses to poxvirus infection.

As per our prior responses, NGS analysis would be technically challenging to perform yet would add little or nothing to our understanding given the inability to address qualitative differences in how different DNA species may serve as agonists for sensors. Moreover, our focus is not on comparing DNA species but on revealing the unexpectedly important role of mtDNA in activating host responses to poxvirus infection.

With regards to controls for cytosol purification, we have modified the text in the figure legend for Supplemental Fig. 5c to make it clearer to readers that samples used for vDNA analysis are

the same as those used for mtDNA analysis in Figure 2a,b, wherein cytosolic controls are presented (namely lack of GAPDH in these fractions). Moreover, for imaging analysis of vDNA in viral factories, these are clearly cytosolic structures.

The experiment demonstrating the depletion of mtDNA following EtBr treatment is not convincing. A 48-hour incubation is not sufficient to observe the depletion of mtDNA. Furthermore, no experiment is proposed to show the amount of mtDNA in the cytosol following EtBr treatment.

While longer treatment periods using lower concentrations of EtBr are often used to eliminate mtDNA, many studies use 48-hour incubations with higher concentrations such as those used here in order to deplete mtDNA from cells more transiently. This is preferable in many cases due to the broader effects of EtBr on genomic DNA. While its efficacy was abundantly clear in the magnitude of the effects of EtBr treatment on host responses to infection in the original submission, we have added new data (Supplementary Fig. 7A of the revised manuscript) further showing that 48-hour treatment with EtBr causes a large reduction in the level of mtDNA nucleoids in cells prior to infection. However, while efficacious and conducted at the Reviewer's request, as we stated in our prior responses, EtBr indiscriminately intercalates into diverse DNA species and has broad effects. As such, we cannot make firm conclusions about the role of mtDNA from EtBr-based approaches, which is why we focus on more specific strategies to address this question.

Figure 2C and 3C: It is difficult for the reviewer to understand how the authors could leave a figure unfinalized.

We respectfully point out that neither of these figures are unfinalized. This is common practice for specific controls, and the panels that the Reviewer refers to simply include controls that serve to show that we can detect apoptosis as we are showing a null effect during infection (Fig. 2C) and that the lack of detectable cGAS in cGAS KO cells is true, and not simply due to failure of the Western blot itself (Fig. 3C).

Reviewer #3 (Remarks to the Author):

I would like to thank the authors for their efforts to address my previous concerns and suggestions.

In my opinion this revision version of the manuscript is easier to read and understand.

In particular, I appreciate the discussion around the recently reported degradation of cGAS by the E5 protein. It was also quite shocking for me and I thought it was worth highlighting in the discussion.

Also, the authors have extended their findings to other cell type (Thp1 monocytes) and additional times after infection.

We thank the Reviewer for their kind and supportive comments.